# Representational Alignment Across Model Layers and Brain Regions with Multi-Level Optimal Transport

**Shaan Shah**
University of California San Diego

**Meenakshi Khosla**
University of California San Diego

## Abstract

Standard representational similarity methods align each layer of a network to its best match in another independently, producing asymmetric results, lacking a global alignment score, and struggling with networks of different depths. These limitations arise from ignoring global activation structure and restricting mappings to rigid one-to-one layer correspondences. We propose Multi-Level Optimal Transport (MOT), a unified framework that jointly infers soft, globally consistent layer-to-layer couplings and neuron-level transport plans. MOT allows source neurons to distribute mass across multiple target layers while minimizing total transport cost under marginal constraints. This yields both a single alignment score for the entire network comparison and a soft transport plan that naturally handles depth mismatches through mass distribution. We evaluate MOT on vision models, large language models, and human visual cortex recordings. Across all domains, MOT matches or surpasses standard pairwise matching in alignment quality. Moreover, it reveals smooth, fine-grained hierarchical correspondences: early layers map to early layers, deeper layers maintain relative positions, and depth mismatches are resolved by distributing representations across multiple layers. These structured patterns emerge naturally from global optimization without being imposed, yet are absent in greedy layer-wise methods. MOT thus enables richer, more interpretable comparisons between representations, particularly when networks differ in architecture or depth. We further extend our method to a three-level MOT framework, providing a proof-of-concept alignment of two networks across their training trajectories and demonstrating that MOT uncovers checkpoint-wise correspondences missed by greedy layer-wise matching.

## 1 Introduction

Understanding high-dimensional neural activity is a shared challenge in neuroscience and artificial intelligence (AI). In neuroscience, comparing neural responses across individuals reveals which computations are universally shared versus idiosyncratic. In AI, comparing representations across models reveals how architectural choices, training objectives, and learning dynamics shape learned features, and helps identify principles of universality i.e. representational properties that emerge consistently across diverse network architectures and objectives. Comparing models to brains extends this logic further: while we cannot rerun biological evolution, we can simulate "evolution in silico" by training artificial networks with different constraints, inputs, and objectives. When such models converge on brain-like representations, they offer mechanistic hypotheses for why the brain may have adopted its computational strategies which is a deeply important theoretical question. These comparisons have revealed striking similarities between biological and artificial networks (Yamins et al. (2014); Eickenberg et al. (2017); Güçlü and Van Gerven (2015); Cichy et al. (2016); Khaligh-Razavi and Kriegeskorte (2014); Schrimpf et al. (2018; 2020); Storrs et al. (2021); Kell et al. (2018)), common computational motifs across diverse architectures and objectives Huh et al. (2024); Kornblith et al. (2019); Bansal et al. (2021); Dravid et al. (2023), and other universal representational dimensions Chen and Bonner (2025); Hosseini et al. (2024).

The standard approach to representational comparison is layer-wise matching: each source layer is paired with the single best-matched target layer under some similarity measure (e.g., Representational

Similarity Analysis (Kriegeskorte et al., 2008), Centered Kernel Alignment (Kornblith et al., 2019), Procrustes distance (Williams et al., 2021) or linear predictivity). Despite its widespread use, this approach has fundamental limitations. It enforces rigid one-to-one correspondences that fail when networks differ in depth or when a source layer corresponds to features distributed across multiple target layers. It produces asymmetric layer mappings depending on the direction of comparison and yields no unified score for global network alignment. Most importantly, by optimizing each match independently, it ignores the global activation structure and risks overfitting to noise.

We propose Multi-Level Optimal Transport (MOT), a framework for globally consistent representational alignment. Optimal transport–based methods, such as Soft-Matching distance (Khosla and Williams, 2024; Khosla et al., 2024), have recently emerged as powerful metrics for comparing neural representations. Unlike metrics such as RSA, CKA, or linear predictivity—which are rotation-invariant and thus unable to capture similarities in neuron-level tuning—OT-based methods are rotation-sensitive. They explicitly match neurons based on their tuning profiles and, by relaxing hard permutation constraints into fractional couplings, can also handle layers of unequal size. This enables richer, more flexible neuron-level alignments than either rotation-invariant similarity metrics or strict permutation-based approaches. However, Soft Matching also remains limited to pairwise layer comparisons like other methods and does not capture global structure across networks. MOT fills this gap by operating across multiple levels: it simultaneously infers soft neuron-to-neuron couplings within layers and a soft, globally consistent layer-to-layer coupling across the two levels. Rather than forcing each source layer to match exactly one target layer, MOT allows source layers to distribute their representational "mass" across multiple target layers while minimizing the total transport cost under marginal constraints; that is, each source layer must distribute exactly 100% of its mass across target layers (no information is lost), and the total mass each target layer receives from all source layers must sum to a balanced allocation (no target is over- or under-utilized). These conservation laws ensure a balanced alignment where every layer contributes meaningfully to the global correspondence, preventing any layer from being arbitrarily overweighted or ignored in the global matching. The result is a single network-level alignment score and a soft transport plan that naturally handles depth mismatches.

We evaluate MOT on three diverse domains: comparisons between foundation models in vision (Vision Transformers like DINOv2 and ViT-MAE), large language models of varying scales (LLaMA, Qwen), and fMRI recordings from human visual cortex across different participants. Our key contributions are:

- **A principled global alignment framework (theoretical):** MOT jointly optimizes all layer correspondences to produce symmetric, globally consistent assignments with a single alignment score. In contrast, greedy pairwise approaches are asymmetric, can overweight certain layers while completely ignoring others, and may spuriously treat wide layers as similar to every layer they are compared against, since their high dimensionality allows them to fit or partially overlap with many different representational subspaces, obscuring more meaningful correspondences.

- **Natural handling of depth mismatches (theoretical):** By allowing soft, many-to-many mappings between layers, MOT aligns networks of different depths without forcing inappropriate one-to-one correspondences.

- **Rotation-invariant extension (theoretical):** We propose an extension of MOT that incorporates additional orthogonal transformations, making the framework rotation-invariant. This ensures that correspondences can be recovered even when shared representational features are embedded in rotated subspaces, and yields consistently high-quality alignments.

- **Improved alignment scores (empirical):** Across domains (vision models, large language models, and brain data), MOT matches or surpasses standard pairwise methods, yielding higher alignment.

- **Emergent hierarchical structure (empirical):** Without imposing ordering constraints, MOT recovers known hierarchical organization in visual cortex data across subjects and reveals clean layer-to-layer correspondences in model–model comparisons where early layers map to early layers and deeper layers preserve their relative ordering. By contrast, greedy pairwise methods fail to reveal hierarchical structure, often leaving many layers unmatched while a single or few layers dominate the mappings.

- **Featural distribution across depth (empirical):** MOT reveals how deeper networks spread computations across multiple layers that shallower networks compress into fewer stages. Concretely, a single layer in a shallower network often distributes its mass across several neighboring layers in a deeper network, an effect that greedy pairwise methods completely miss.

## 2 METHODS

### 2.1 PROBLEM SETUP AND EXISTING APPROACHES

Comparing the internal representations of neural networks often proceeds by a *pairwise* layer search under some similarity or distance measure. In general, let $T$ denote the number of stimuli (e.g., images, text sequences) used to probe both models. We consider two neural networks:

- The first network has $L$ layers. Layer $\ell$ contains $n_\ell$ units, and its activations across the $T$ stimuli are represented by

$$X_\ell \in \mathbb{R}^{T \times n_\ell}, \quad \ell = 1, \dots, L.$$

- The second network has $M$ layers. Layer $m$ contains $n_m$ units, with activations

$$Y_m \in \mathbb{R}^{T \times n_m}, \quad m = 1, \dots, M.$$

Each row of $X_\ell$ or $Y_m$ corresponds to the response of all units in that layer to a single stimulus, while each column corresponds to the activity of a single unit across stimuli. For any chosen alignment metric $S(\cdot, \cdot)$ (e.g., linear predictivity, Procrustes distance, RSA, CKA, or Soft Matching), one typically does:

$$m^*(\ell) = \arg \max_m S(X_\ell, Y_m)$$

and then reports the layer-wise score $S(X_\ell, Y_{m^*(\ell)})$. This enforces a hard one-to-one mapping from each source layer $\ell$ to a single target layer $m^*(\ell)$. However, when $L \neq M$, or when the set of features represented in one layer of network A is distributed across multiple layers of network B, such a rigid per-layer pairing is not well-suited: a source layer may genuinely correspond to a mixture of multiple target layers. Moreover, by optimizing each layer independently, this approach ignores the *global* structure of all activations and can overfit to noise in any single layer's responses.

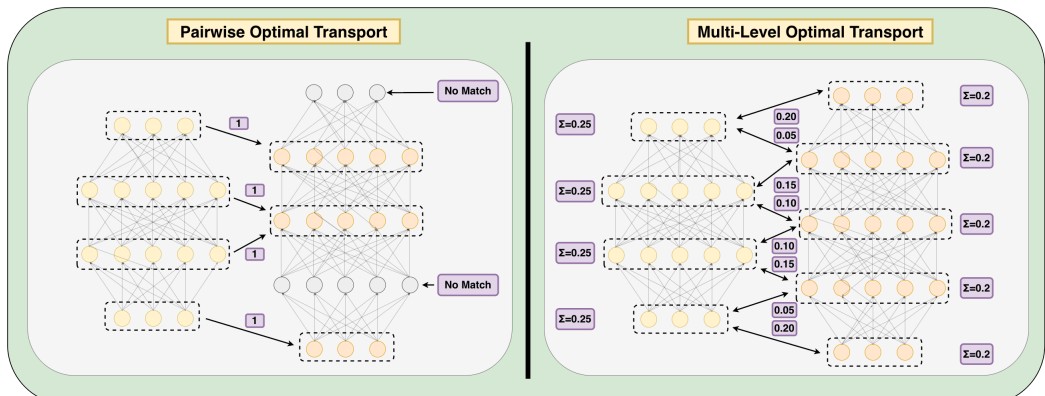

Figure 1: *Left*: **Pairwise OT**. Layers are matched independently, so multiple target layers can be mapped to the same source while other sources remain unused, yielding asymmetric, unbalanced mappings. *Right*: **Multi-Level OT**. MOT infers a globally consistent transport plan where each source layer distributes all its mass and each target layer receives exactly one unit, ensuring balanced, symmetric alignments and handling depth mismatches.

### 2.2 MULTI-LEVEL OPTIMAL TRANSPORT

Our framework operates at two levels: an inner level that aligns neurons within each layer pair, and an outer level that determines how layers should be coupled globally.

### 2.2.1 INNER LEVEL: NEURON-TO-NEURON TRANSPORT

For each candidate layer pair $(\ell, m)$, MOT computes a neuron-level alignment cost by applying the soft-matching OT (Appendix A for more details) formulation to the tuning functions in $X_\ell$ and $Y_m$. We first construct a pairwise dissimilarity matrix

$$C_{\ell m}^{\text{inner}}[i, j] = c(X_\ell[:, i], Y_m[:, j]),$$

where $c(\cdot, \cdot)$ is the chosen tuning-based dissimilarity (here, correlation distance). The inner OT objective then computes

$$C_{\ell m} = \min_{Q_{\ell m} \in \mathcal{T}(n_\ell, n_m')} \langle C_{\ell m}^{\text{inner}}, Q_{\ell m} \rangle,$$

yielding a soft coupling $Q_{\ell m}$ that specifies how strongly each neuron in layer $\ell$ corresponds to neurons in layer $m$.

Because $Q_{\ell m}$ lies in the transportation polytope, its row and column marginals ensure that each source neuron distributes its representational mass uniformly and each target neuron receives its appropriate share. When the two layers have the same width ($n_\ell = n_m'$), the optimal solution reduces to a permutation (by the Birkhoff–von Neumann theorem (von Neumann, 1953; De Loera and Kim, 2013)); when they differ, soft assignments naturally emerge, allowing neurons in one layer to map to mixtures of neurons in the other. The resulting inner OT cost $C_{\ell m}$ serves as the layer-to-layer dissimilarity used by the outer level of MOT.

### 2.2.2 OUTER LEVEL: LAYER-TO-LAYER TRANSPORT

The inner costs $C_{\ell m}$ form a layer-to-layer cost matrix $C \in \mathbb{R}^{L \times M}$. We solve a second optimal transport problem to find the global layer coupling:

$$P = \arg \min_{P \in \mathcal{T}(L, M)} \langle C, P \rangle$$

where $\mathcal{T}(L, M)$ is the transportation polytope for layers:

$$\mathcal{T}(L, M) = \left\{ P \in \mathbb{R}^{L \times M} : \sum_\ell P_{\ell m} = \tfrac{1}{M}, \ \sum_m P_{\ell m} = \tfrac{1}{L}, \ P_{\ell m} \geq 0 \right\}.$$

The element $P_{\ell m}$ represents the fraction of layer $\ell$'s representation that is explained by layer $m$. These marginal constraints ensure mass conservation: each source layer distributes its unit mass across target layers (after normalization by $L$), and the total mass received by all target layers is balanced.

When the two networks have the same number of layers ($L = M$), we can show that the optimal solution $P$ is a (scaled) permutation matrix: the objective $\langle C, P \rangle$ is linear in $P$, and the constraints define the transportation polytope whose vertices are permutation matrices (scaled by $1/L$) by the Birkhoff–von Neumann theorem. Since linear programs achieve their optima at vertices, the solution finds a one-to-one layer matching when $L = M$. However, when $L \neq M$, the soft coupling allows source layers to distribute their mass across multiple target layers, naturally handling depth mismatches.

### 2.2.3 RECONSTRUCTION AND EVALUATION

Given the layer coupling $P$ and neuron couplings $\{Q_{\ell m}\}$, we reconstruct layer $\ell$ as:

$$\hat{X}_\ell = L \sum_{m=1}^{M} P_{\ell m} Y_m Q_{\ell m}^\top.$$

The factor $L$ appears because $P_{\ell m}$ sums to $1/L$ across $m$. We evaluate alignment quality using the mean correlation between original and reconstructed neurons on held-out data:

$$\text{Score}_\ell = \frac{1}{n_\ell} \sum_{i=1}^{n_\ell} \rho(X_\ell[:, i], \hat{X}_\ell[:, i]).$$

The global MOT score is the average across all layers:

$$\text{MOT} = \frac{1}{L} \sum_\ell \text{Score}_\ell.$$

## 2.3 Rotation-Invariant Extension

Most existing metrics of representational similarity, such as RSA, CKA, and Procrustes distance, are designed to be rotation-invariant. The rationale is that when comparing population codes, we often care less about the tuning of individual neurons and more about the geometry of the representational space: distances, angles, and relative positions between stimulus responses. Two networks can encode essentially the same geometry while using different coordinate bases (for example, rotated versions of one another). Requiring strict unit-to-unit correspondence in such cases would artificially inflate dissimilarity, even though the representational geometry and information content are preserved. A rotation-invariant extension of MOT (**MOT + R** henceforth) therefore provides a way to capture this geometric equivalence while still enforcing a globally consistent layer- and neuron-level coupling. We introduce rotation matrices $R_{\ell m} \in O(n_\ell)$ for each layer pair and minimize the reconstruction error:

$$C_{\ell m} = \min_{Q_{\ell m}, R_{\ell m}} \|X_\ell R_{\ell m} - Y_m Q_{\ell m}^\top\|_F^2.$$

We optimize via alternating minimization:

- **Fix $R$, update $Q$:** Solve optimal transport using correlation distance on rotated features $X_\ell R_{\ell m}$.
- **Fix $Q$, update $R$:** Solve orthogonal Procrustes via SVD of $X_\ell^\top (Y_m Q_{\ell m}^\top)$.
- **Update $P$:** Refresh the outer coupling using the Frobenius reconstruction costs.

Predictions incorporate the learned rotations:

$$\hat{X}_\ell = L \sum_{m=1}^{M} P_{\ell m} Y_m Q_{\ell m}^\top R_{\ell m}^\top.$$

## 2.4 Baseline Comparisons

We compare MOT against several baselines to isolate the contribution of each component:

**Random Layer Assignment (Perm-P):** We randomly permute the rows of $P$, breaking the optimized layer correspondences while preserving the neuron-level optimal transport within each layer pair. This control is expected to perform reasonably well because it still finds optimal neuron matches for each (now randomly assigned) layer pairing—it only disrupts which layers are matched, not how well neurons align within those matches.

**MOT Top-1 Layer Transport Plan (Single-Best OT):** Each source layer $\ell$ maps only to its highest-weight target layer

$$m^*(\ell) = \arg\max_m P_{\ell m},$$

converting the soft layer coupling to a hard one-to-one assignment while keeping the soft neuron-level transport.

**Independent Pairwise OT (Pairwise Best OT):** The standard greedy approach where each source layer is matched to the single target layer with minimum inner OT cost, computed independently on training data. Formally,

$$m^*(\ell) = \arg\min_m C_{\ell m},$$

and layer $\ell$ is reconstructed using only $Y_{m^*(\ell)}$. This baseline ignores global structure and can result in multiple source layers matching to the same target layer while leaving others unmatched.

**Rotation-aware variants:** We evaluate rotation-invariant versions of the pairwise OT baseline (henceforth, 'Pairwise Best + R'), where predictions use $Y_m Q_{\ell m}^\top R_{\ell m}^\top$ with orthogonal transformations optimized via Procrustes alignment.

## 3 RESULTS

We evaluate representational similarity under the MOT metric across four distinct alignment setups: (i) large language models of different families and scales, (ii) fMRI responses from the visual cortex of four human subjects, (iii) pretrained transformer-based vision models spanning different families and scales, (iv) cross-domain comparisons between human visual cortex and vision transformers (Appendix Section E). Across all settings, MOT matches or surpasses baseline reconstruction (prediction) scores. Importantly, the transport plans inferred by MOT naturally reveal systematic layer-wise correspondences across models and cortex, despite not being explicitly optimized for such structure; specifically, earlier layers or regions tend to align with earlier counterparts, while deeper layers or regions map to progressively higher levels; patterns that greedy pairwise methods fail to capture. Furthermore, we demonstrate a proof-of-concept extension of MOT to a Three-level MOT framework by aligning two networks across their training trajectories.

### 3.1 REPRESENTATIONAL SIMILARITY BETWEEN LARGE LANGUAGE MODELS

**Experimental Setup.** We extract layer-wise representations by averaging token activations across 2,552 prompts from the STSB dataset (May, 2021; Enevoldsen et al., 2025; Muennighoff et al., 2022). For each model, this yields a sequence of representation matrices $X$ and $Y$, corresponding to successive layers. We evaluate models of varying sizes from the LLaMA-3.2 (Grattafiori et al., 2024) and Qwen-2.5 (Qwen et al., 2025) families. Representations are compared using MOT and baseline metrics, and the resulting transport plans are analyzed. To quantify alignment quality, we reconstruct representations on a held-out validation split (20%) using the learned transport maps and report the correlation with ground-truth activations, as detailed in the Methods Section2.2.3.

| Model 1 | Model 2 | MOT Metric | Random (Perm-P) | Single-Best OT | Pairwise Best OT |
|---------|---------|------------|-----------------|----------------|------------------|
| Llama-3.2 1B | Llama-3.2 3B | **0.558** | 0.510 | 0.502 | 0.505 |
| Qwen-2.5 0.5B | Qwen-2.5 3B | **0.510** | 0.494 | 0.467 | 0.477 |
| Qwen-2.5 0.5B | Llama-3.2 1B | **0.522** | 0.500 | 0.502 | 0.511 |
| Qwen-2.5 0.5B | Llama-3.2 3B | **0.531** | 0.513 | 0.498 | 0.524 |
| Llama-3.2 1B | Qwen-2.5 3B | **0.432** | 0.411 | 0.345 | 0.380 |
| Llama-3.2 3B | Qwen-2.5 3B | **0.383** | 0.374 | 0.338 | 0.346 |

Table 1: **LLM alignment performance**. Comparison of MOT against baseline metrics, evaluated by reconstruction correlation on held-out data.

**Results.** MOT consistently achieves higher reconstruction accuracy than baseline methods. As shown in Table 1, reconstruction scores on the held-out validation set are significantly higher under MOT, indicating that the alignment plans it learns are more robust and generalizable than those obtained from pairwise baselines. This improvement stems from MOT's (i) enforcement of global consistency across all layers and (ii) ability to distribute representational mass across multiple target layers, allowing it to recover alignments that remain hidden to pairwise methods when features from a single layer are distributed across several layers in another model. Beyond quantitative gains, MOT uncovers clear hierarchical correspondences between models. As illustrated in Figures 2 and B.1, the transport plans produced by MOT exhibit strong diagonal structure: early layers in one model align with early layers in the other, while deeper layers align with deeper layers. Such structured correspondences are absent in pairwise OT, which often yields noisy mappings. Furthermore, when comparing shallower with deeper models, MOT reveals that single layers in the shallower model distribute their mass across multiple consecutive layers in the deeper model. This soft many-to-many mapping reflects how additional depth refines and spreads computations, suggesting that representational stages compressed into one layer in a shallower model are decomposed across multiple processing steps in a deeper model—an organizational principle that greedy baselines fail to uncover.

To rule out the possibility that this effect stems from the multi-level formulation and not from OT, we also perform pairwise linear predictivity and RSA analyses. As shown in Figures H.1 and I.1, these methods also fail to reveal hierarchial correspondence.

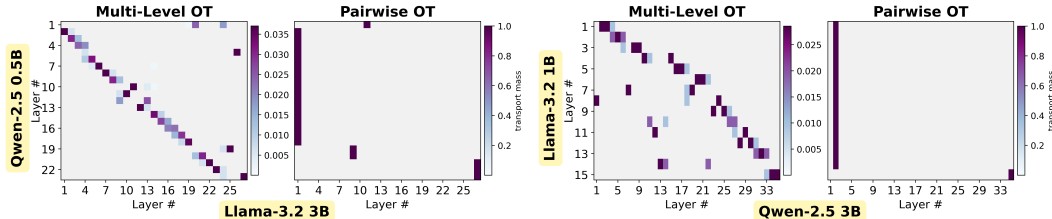

Figure 2: **Transport plans for LLM alignment.** Multi-Level OT (left) versus pairwise OT (right) for two cross-model comparisons: (a) Qwen-2.5 0.5B ↔ LLaMA-3.2 3B and (b) LLaMA-3.2 1B ↔ Qwen-2.5 3B. MOT reveals smooth, diagonal correspondences across layers, while pairwise OT produces noisier and less structured mappings.

## 3.2 Representational Similarity Between Visual Cortex across Subjects

**Experimental Setup.** We analyze fMRI responses from the Natural Scenes Dataset (NSD; (Allen et al., 2022)), which contains recordings from 8 participants who each viewed up to 10,000 natural images. Of these, 4 subjects viewed the full set of 10,000 images three times, with 1,000 images shared across all participants. We focus on these 4 subjects and restrict our analysis to their responses to the shared 1,000 images, ensuring that alignment is evaluated on common stimuli.

We target visual cortex regions (V1–V4, Lateral, Dorsal, Ventral), treating each region as a "layer" and individual voxels as "neurons". The visual cortex provides a strong testbed for multi-level alignment because it is one of the best-characterized cortical systems: early areas (V1, V2) are known to encode low-level features such as orientation and contrast, while higher areas (V3, V4) encode progressively more complex shapes and object features (Hubel and Wiesel, 1968; Pasupathy and Connor, 2001; Desimone and Schein, 1987; Desimone et al., 1984). This hierarchical progression is well-established across individuals, so correspondences between homologous regions are strongly expected. MOT is applied to align responses across subjects by generating transport maps between regions. To evaluate alignment quality, we reconstruct held-out responses (20% validation split) and report correlations with ground-truth activity. We repeat this evaluation across 5 random train-validation splits to ensure robustness.

| Model 1 | Model 2 | MOT Metric | Random (Perm-P) | Single-Best OT | Pairwise Best OT |
|---------|---------|------------|-----------------|----------------|------------------|
| Subject A | Subject B | 0.244 | 0.135 | 0.244 | **0.245** |
| Subject A | Subject C | 0.199 | 0.110 | 0.199 | **0.202** |
| Subject A | Subject D | 0.198 | 0.109 | 0.198 | **0.201** |
| Subject B | Subject C | **0.212** | 0.126 | **0.212** | 0.212 |
| Subject C | Subject D | 0.197 | 0.112 | 0.197 | **0.199** |
| Subject B | Subject D | 0.201 | 0.121 | 0.201 | **0.204** |

Table 2: **Visual cortex alignment performance.** Comparison of MOT against baseline methods, evaluated by reconstruction correlation on held-out fMRI responses.

**Results.** As shown in Tables 2 and C.1, MOT achieves reconstruction scores comparable to pairwise OT, with only a marginal decrease in correlation. More importantly, Figures 3 and C.1 show that the transport maps inferred by MOT recover the expected cross-subject correspondences: cortical regions in one subject consistently align to the same regions in another. In contrast, pairwise OT does not produce such structured mappings in any subject pair. This indicates that MOT's global alignment scheme captures region-to-region correspondences that pairwise methods miss. We further assess pairwise linear predictivity and RSA, with the results as shown in Figures H.1 and I.1. The layers selected as best-matching by these approaches fail to align corresponding regions as well, reinforcing that the multi-level alignment mechanism in MOT is crucial for obtaining robust and generalizable correspondences. Overall, these results demonstrate that MOT yields transport plans that are both interpretable and biologically meaningful, while maintaining reconstruction performance on par with baseline methods.

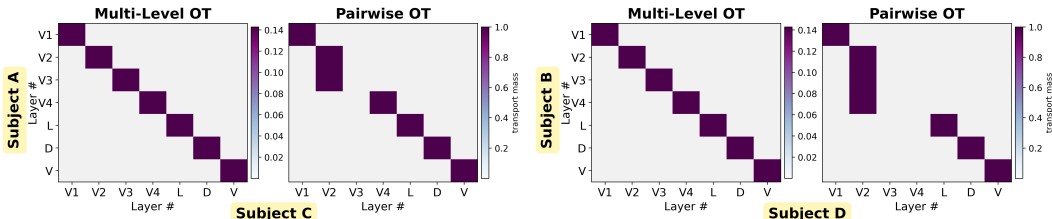

Figure 3: **Transport plans for cross-subject brain alignment.** Multi-Level OT (left) versus pairwise OT (right) for two randomly selected subject pairs: (a) Subject A ↔ Subject C and (b) Subject B ↔ Subject D. MOT recovers structured region-to-region correspondences that are absent in pairwise OT. Other subject pairs show similar trends (see Appendix C.1).

## 3.3 REPRESENTATION SIMILARITY BETWEEN VISION MODELS

**Experimental Setup.** We extract layer-wise representations from Vision Transformers by averaging patch activations for each input image. We use 20,000 images randomly sampled from the ImageNet validation set  (Deng et al., 2009; Russakovsky et al., 2015), sampled to ensure a uniform coverage across all classes in Imagenet. For each model, this yields a sequence of representation matrices that serve as inputs to the MOT framework. We evaluate two families of pretrained vision transformers—DINOv2 and ViT-MAE across multiple model scales. Alignment quality is quantified by reconstructing representations on a held-out validation split (20%) using the learned transport plans, with reconstruction–ground truth correlation as the metric.

Prior work has shown that the residual stream in Transformers lacks privileged axes and is invariant up to rotations of its basis (Khosla et al., 2024). Since MOT, like other OT-based methods, is rotation-sensitive, we additionally evaluate a rotation-augmented variant (MOT+R) and its baselines (see Methods Section 2.3). We restrict this analysis to Vision Transformers, where rotational invariances are especially relevant and where the computational cost of MOT+R remains tractable. For large language models and fMRI data, the added optimization over rotations was computationally prohibitive, so we report only the rotation-sensitive results in those domains. In MOT+R, the learned rotation matrices are incorporated into both transport optimization and evaluation, ensuring that geometric equivalences induced by rotations are properly captured.

| Model 1 | Model 2 | MOT Metric | Pairwise Best OT | MOT + R | Pairwise Best + R |
|---------|---------|------------|------------------|---------|-------------------|
| DINOv2 Small | ViT-MAE Base | 0.289 | 0.301 | **0.600** | 0.526 |
| DINOv2 Small | DINOv2 Large | 0.353 | 0.340 | **0.778** | 0.394 |
| DINOv2 Small | DINOv2 Giant | 0.466 | 0.433 | **0.790** | 0.418 |
| DINOv2 Small | ViT-MAE Large | 0.381 | 0.354 | **0.633** | 0.509 |
| DINOv2 Small | ViT-MAE Huge | 0.411 | 0.386 | **0.657** | 0.508 |
| ViT-MAE Base | DINOv2 Large | 0.577 | 0.624 | **0.732** | 0.283 |
| ViT-MAE Base | DINOv2 Giant | 0.202 | 0.180 | **0.580** | 0.293 |
| ViT-MAE Base | ViT-MAE Large | 0.588 | 0.598 | **0.850** | 0.596 |
| ViT-MAE Base | ViT-MAE Huge | 0.149 | 0.417 | **0.788** | 0.571 |
| ViT-MAE Huge | DINOv2 Giant | 0.317 | 0.352 | **0.614** | 0.359 |

Table 3: **Vision model alignment performance** Reconstruction accuracy under MOT and pairwise OT, reported both in the standard (rotation-sensitive) and rotation-augmented (MOT+R) settings.

**Results.** Table 3 shows that, on its own, vanilla MOT does not consistently outperform pairwise OT in reconstruction accuracy. The transport plans inferred by MOT (Figure  4 and Figures  D.1 -  D.10) only partially reveal layer-wise correspondences: in some cases, clear diagonal structure emerges, but in others the mappings are noisier. By contrast, the rotation-augmented variant (MOT+R) yields substantially higher reconstruction scores surpassing both vanilla MOT and pairwise vanilla and rotational baselines (Tables  3,  D.1 and  D.2). Importantly, the transport plans produced by MOT+R consistently exhibit strong hierarchical correspondences, recovering clean layer-to-layer alignment even in settings where vanilla MOT fails to do so. These findings indicate that incorporating rotation

into the OT framework not only improves quantitative alignment quality but also produces more generalizable and interpretable mappings, particularly in domains like Vision Transformers where representations are known to be rotation-invariant.

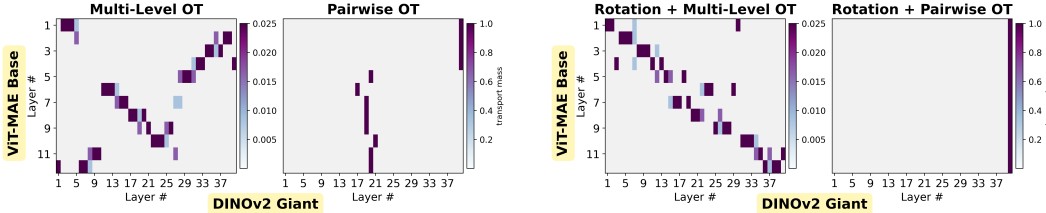

Figure 4: **Transport plans for vision model alignment.** ViT-MAE Base $\leftrightarrow$ DINOv2 Giant (a) without rotation (MOT) and (b) with rotation augmentation (MOT+R). MOT+R captures geometric equivalences induced by rotations, yielding clearer correspondences than the rotation-sensitive variant.

### 3.4 THREE-LEVEL MOT FOR ALIGNING TRAINING TRAJECTORIES

**Formulation of Three-Level MOT**. We extend the two-level MOT formulation described above to an additional level over checkpoints. Let the first model have checkpoints $c = 1, \ldots, C_A$ and the second $d = 1, \ldots, C_B$, with checkpoint-$c$ activations $X_\ell^{(c)}$ and checkpoint-$d$ activations $Y_m^{(d)}$.

For each checkpoint pair $(c, d)$, we treat $X_\ell^{(c)}$ and $Y_m^{(d)}$ as two networks and run the same two-level MOT procedure as above, yielding neuron- and layer-level couplings together with a single scalar cost

$$C_{cd}^{\text{chkpt}} = \text{MOT}\big(X_\ell^{(c)}; Y_m^{(d)}\big),$$

where $\text{MOT}(\cdot, \cdot)$ denotes the two-level objective (inner neuron OT followed by outer layer OT). This defines a checkpoint-level cost matrix $C^{\text{chkpt}} \in \mathbb{R}^{C_A \times C_B}$. To obtain a globally consistent alignment between the two training trajectories, we then solve a third OT problem over checkpoints, reusing the same uniform-marginal formulation as before:

$$R = \arg\min_{R \in \mathcal{T}(C_A, C_B)} \langle C^{\text{chkpt}}, R \rangle,$$

and $R_{cd}$ can be interpreted as a soft assignment between checkpoint $c$ in the first model and checkpoint $d$ in the second. For comparison, we also construct a simple pairwise-best checkpoint map that ignores the global OT constraint and matches each checkpoint independently to its lowest-cost partner.

**Experimental Setup and Results**. We perform three-level OT (and the corresponding pairwise baseline) as a proof of concept for 154 checkpoints of Pythia-14M models with seed 1 and seed 2 (Biderman et al. (2023)), where the 154 checkpoints correspond to steps 0 (initialization), 1, 2, 4, 8, 16, 32, 64, 128, 256, 512, 1000, and then every 1,000 subsequent steps. The layer-wise representations for each checkpoint are collected across 2,552 prompts from the STSB dataset (May, 2021; Enevoldsen et al., 2025; Muennighoff et al., 2022). We plot the checkpoint-level transport maps generated by three-level MOT and by the pairwise-best alignment in Figure 5. MOT reveals a coherent checkpoint-wise correspondence between the two training trajectories, whereas pairwise matching fails to do so. This demonstrates a proof-of-concept method for aligning entire training trajectories of two models. In principle, the MOT framework can be generalized to any number of levels, depending on the type of alignment setting being studied.

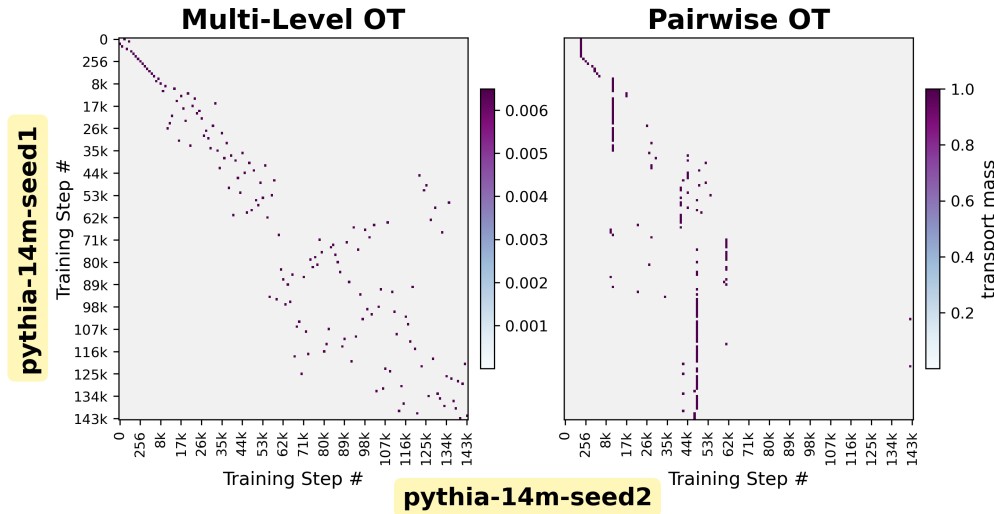

Figure 5: **Transport plans for training-trajectory alignment.** Three-level MOT (left) and pairwise OT (right) for Pythia-14M-seed1 $\leftrightarrow$ Pythia-14M-seed2 across training checkpoints. MOT yields an, approximately diagonal checkpoint correspondence, whereas pairwise OT shows no clear trajectory-level alignment.

## 4  DISCUSSION

Multi-Level OT provides a principled framework for aligning two networks with arbitrary depths. By softly coupling all layer objectives, MOT (i) allows a neuron in one layer of a source network to align with a soft combination of units distributed across multiple layers of a target network, (ii) enforces global consistency that mitigates overfitting to noise in any single layer, and (iii) produces a single network-level—rather than layer-level—alignment score. In our experiments, MOT achieves higher alignment scores than greedy pairwise matches and yields intuitive transport plans that reveal hierarchical correspondences both in artificial networks and in the human brain.

**Limitations.**   Despite these strengths, several limitations remain. First, MOT is computationally demanding: solving the inner OT scales as $O(n^3 \log n)$ for $n$ neurons in a pair of layers, and the overall procedure is $O(L^2 n^3 \log n)$ for networks with $L$ layers. This makes scaling to very wide or deep models challenging without further algorithmic improvements. Second, our evaluation was limited to a subset of models and brain datasets; follow-up analyses on more models and diverse neural data will be critical to assess generality. Finally, our focus here is on developing a principled method for quantifying representational alignment, rather than explaining why different systems converge on similar representations. The latter is a separate and substantially harder theoretical question. Recent ideas, such as the contravariance principle (which frames convergence as a consequence of shared task or ecological constraints) (Cao and Yamins, 2024) and the Platonic representation hypothesis (which proposes that training on rich naturalistic data drives different systems toward similar underlying structure in the world) (Huh et al., 2024), offer intriguing conceptual directions but remain challenging to evaluate empirically. By providing a rigorous way to characterize alignment, MOT offers the measurement foundation needed for such explanatory hypotheses to be tested in future work.

**Future directions.**   Several extensions are natural. One is to study training-run level alignment in more depth, examining how model architecture, training data order, learning rate, and other optimization choices shape the resulting alignment plan. Another is to incorporate priors on the transport plan (e.g. smoothness) to guide alignment toward more interpretable solutions. Ultimately, as both biological and artificial networks grow in scale and complexity, methods like MOT that respect global structure while revealing network correspondences will be essential for understanding the universal principles governing intelligent systems.

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

APPENDIX

## A  OPTIMAL TRANSPORT FOR REPRESENTATIONAL ALIGNMENT

Optimal Transport (OT) provides a principled way to compare two neural populations by framing each representation as a probability distribution over its tuning functions and measuring the minimal "effort" required to transform one representational distribution into another. Concretely, if $X \in \mathbb{R}^{T \times N_x}$ and $Y \in \mathbb{R}^{T \times N_y}$ denote two sets of tuning curves across $T$ stimuli, we represent each population as a uniform mixture of Dirac masses over its neurons:

$$\mu_X = \frac{1}{N_x} \sum_{i=1}^{N_x} \delta_{x_i}, \qquad \mu_Y = \frac{1}{N_y} \sum_{j=1}^{N_y} \delta_{y_j}.$$

The cost function $C_{ij} = c(x_i, y_j)$ quantifies the dissimilarity between the tuning functions of neuron $i$ in $X$ and neuron $j$ in $Y$; depending on the application, $c(\cdot, \cdot)$ may be squared Euclidean distance, correlation distance, or any appropriate tuning-based dissimilarity. The 2-Wasserstein or the soft-matching (Khosla and Williams (2024)) distance between $\mu_X$ and $\mu_Y$ is

$$d_T(X, Y) = \min_{P \in \mathcal{T}(N_x, N_y)} \sum_{ij} P_{ij} \, C_{ij},$$

where $P$ ranges over the *transportation polytope* $\mathcal{T}(N_x, N_y) = \Big\{ P \in \mathbb{R}^{N_x \times N_y} : P_{ij} \geq 0, \ \sum_i P_{ij} = \frac{1}{N_y}, \ \sum_j P_{ij} = \frac{1}{N_x} \Big\}$. Any $P \in \mathcal{T}(N_x, N_y)$ is a *doubly stochastic coupling*: it assigns a non-negative "mass" $P_{ij}$ to transporting representational weight from neuron $i$ in $X$ to neuron $j$ in $Y$, with uniform row and column marginals. Unlike rotation-invariant metrics such as RSA or CKA, OT preserves single-neuron tuning structure and yields explicit, interpretable neuron-to-neuron correspondences.

MOT builds directly on this OT foundation by elevating the coupling to the level of layers, simultaneously inferring soft layer-to-layer couplings and per-neuron transport plans to produce a globally balanced alignment across networks of arbitrary depths (Figure 1).

# B COMPARING LLM-LLM REPRESENTATIONS

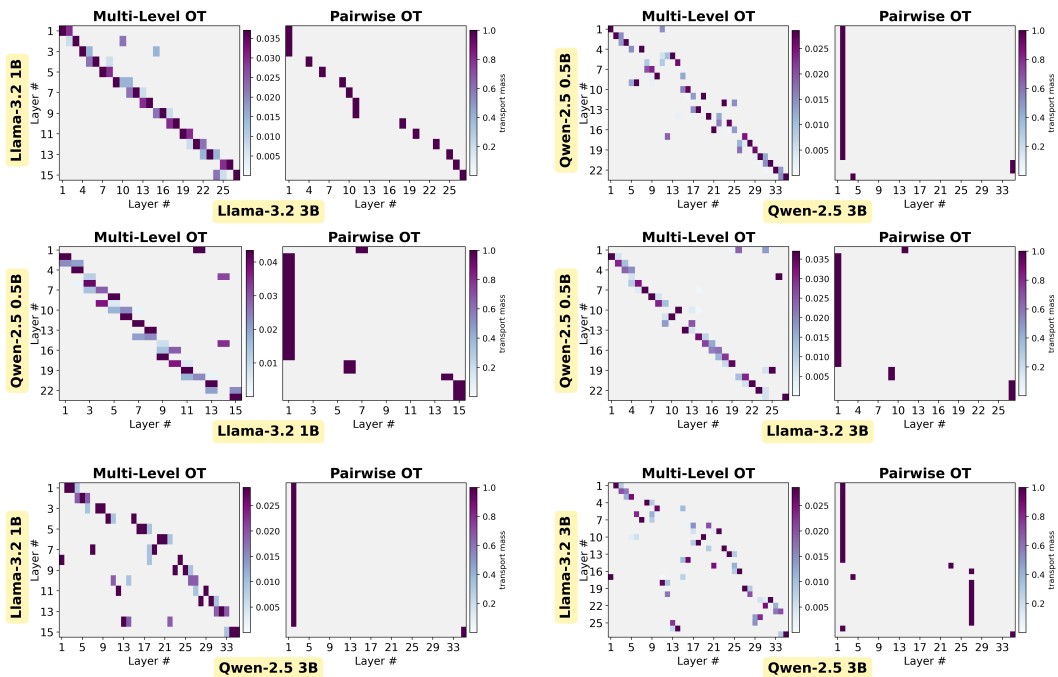

Figure B.1: **Transport plans across LLM families and scales.** Multi-Level OT (MOT) mappings are shown for six cross-model comparisons: (a) LLaMA-3.2 1B ↔ LLaMA-3.2 3B, (b) Qwen-2.5 0.5B ↔ Qwen-2.5 3B, (c) Qwen-2.5 0.5B ↔ LLaMA-3.2 1B, (d) Qwen-2.5 0.5B ↔ LLaMA-3.2 3B, (e) LLaMA-3.2 1B ↔ Qwen-2.5 3B, and (f) LLaMA-3.2 3B ↔ Qwen-2.5 3B. MOT uncovers structured, near-diagonal correspondences that persist across both intra-family (a,b) and cross-family (c-f) alignments, illustrating its robustness compared to pairwise OT.

# C   REPRESENTATION SIMILARITY BETWEEN VISION CORTEX RESPONSE

| Model 1 | Model 2 | MOT Metric | Random (Perm-P) | Single-Best OT | Pairwise Best OT |
|---|---|---|---|---|---|
| Subject A | Subject B | $0.244 \pm 0.005$ | $0.135 \pm 0.015$ | $0.244 \pm 0.005$ | **0.245 ± 0.004** |
| Subject A | Subject C | $0.199 \pm 0.003$ | $0.110 \pm 0.013$ | $0.199 \pm 0.003$ | **0.202 ± 0.003** |
| Subject A | Subject D | $0.198 \pm 0.008$ | $0.109 \pm 0.016$ | $0.198 \pm 0.008$ | **0.201 ± 0.007** |
| Subject B | Subject C | **0.212 ± 0.007** | $0.126 \pm 0.014$ | **0.212 ± 0.007** | **0.212 ± 0.007** |
| Subject C | Subject D | $0.197 \pm 0.007$ | $0.112 \pm 0.016$ | $0.197 \pm 0.007$ | **0.199 ± 0.006** |
| Subject B | Subject D | $0.201 \pm 0.008$ | $0.121 \pm 0.014$ | $0.201 \pm 0.008$ | **0.204 ± 0.007** |

Table C.1: **Visual cortex alignment performance.** Comparison of MOT against baseline methods, evaluated by reconstruction correlation on held-out fMRI responses: mean $\pm$ standard deviation across seeds.

## C.1   TRANSPORT PLANS FOR OT BASED MAPPING PAIRS

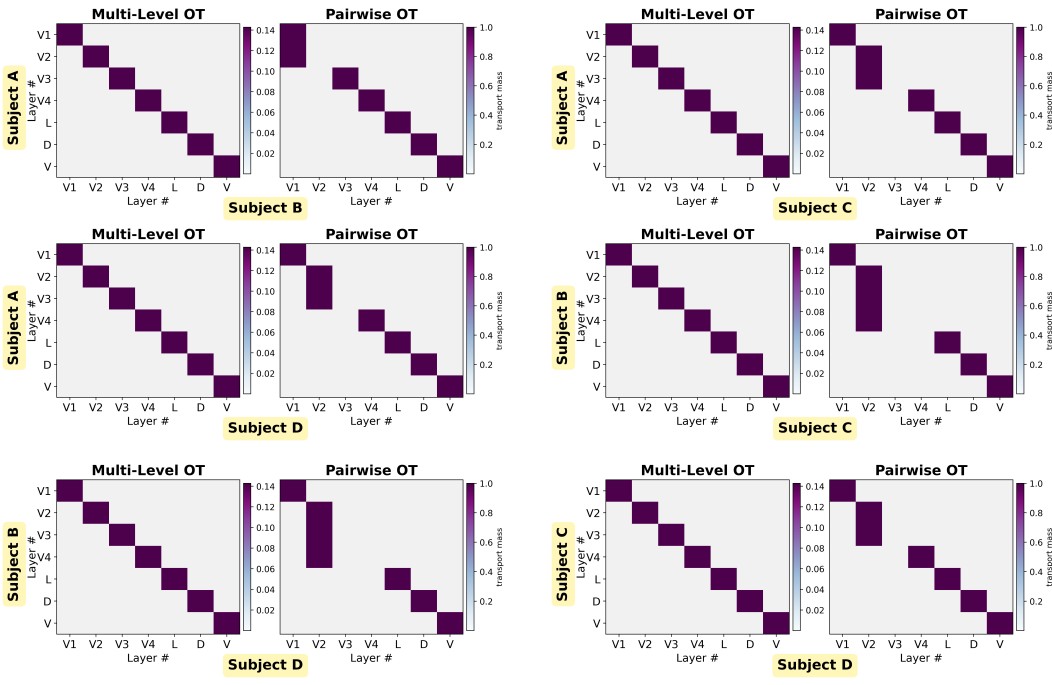

Figure C.1: **Transport plans for OT-based mapping pairs.** Pairwise OT mappings across all subject pairs: (a) Subject A $\leftrightarrow$ Subject B, (b) Subject A $\leftrightarrow$ Subject C, (c) Subject A $\leftrightarrow$ Subject D, (d) Subject B $\leftrightarrow$ Subject C, (e) Subject B $\leftrightarrow$ Subject D, and (f) Subject C $\leftrightarrow$ Subject D. MOT recovers structured region-to-region correspondences that are absent in pairwise OT.

# D    REPRESENTATION SIMILARITY BETWEEN VISION MODELS

| Model 1 | Model 2 | MOT Metric | Random (Perm-P) | Single-Best OT | Pairwise Best OT |
|---|---|---|---|---|---|
| DINOv2 Small | ViT-MAE Base | 0.289 | 0.259 | 0.289 | **0.301** |
| DINOv2 Small | DINOv2 Large | **0.353** | 0.298 | 0.312 | 0.340 |
| DINOv2 Small | DINOv2 Giant | **0.466** | 0.385 | 0.413 | 0.433 |
| DINOv2 Small | ViT-MAE Large | **0.381** | 0.348 | 0.350 | 0.354 |
| DINOv2 Small | ViT-MAE Huge | **0.411** | 0.372 | 0.359 | 0.386 |
| ViT-MAE Base | DINOv2 Large | 0.577 | 0.394 | 0.531 | **0.624** |
| ViT-MAE Base | DINOv2 Giant | **0.202** | 0.197 | 0.148 | 0.180 |
| ViT-MAE Base | ViT-MAE Large | 0.588 | 0.528 | 0.539 | **0.598** |
| ViT-MAE Base | ViT-MAE Huge | 0.149 | 0.116 | 0.181 | **0.417** |
| ViT-MAE Huge | DINOv2 Giant | 0.317 | 0.261 | 0.293 | **0.352** |

Table D.1: MOT Metric with its baseline comparisons.

| Model 1 | Model 2 | MOT + R | Single-Best + R | Pairwise Best + R |
|---|---|---|---|---|
| DINOv2 Small | ViT-MAE Base | **0.600** | **0.600** | 0.526 |
| DINOv2 Small | DINOv2 Large | **0.778** | 0.708 | 0.394 |
| DINOv2 Small | DINOv2 Giant | **0.790** | 0.746 | 0.418 |
| DINOv2 Small | ViT-MAE Large | **0.633** | 0.599 | 0.509 |
| DINOv2 Small | ViT-MAE Huge | **0.657** | 0.614 | 0.508 |
| ViT-MAE Base | DINOv2 Large | **0.732** | 0.712 | 0.283 |
| ViT-MAE Base | DINOv2 Giant | **0.580** | 0.605 | 0.293 |
| ViT-MAE Base | ViT-MAE Large | **0.850** | 0.848 | 0.596 |
| ViT-MAE Base | ViT-MAE Huge | **0.788** | 0.760 | 0.571 |
| ViT-MAE Huge | DINOv2 Giant | **0.614** | 0.582 | 0.359 |

Table D.2: Rotation (MOT + R) with its baseline comparisons.

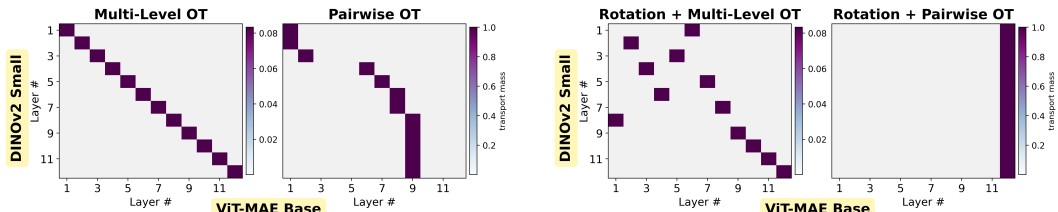

Figure D.1: **Transport plans for vision model alignment.** DINOv2 Small ↔ ViT-MAE Base (a) without rotation (MOT) and (b) with rotation augmentation (MOT+R).

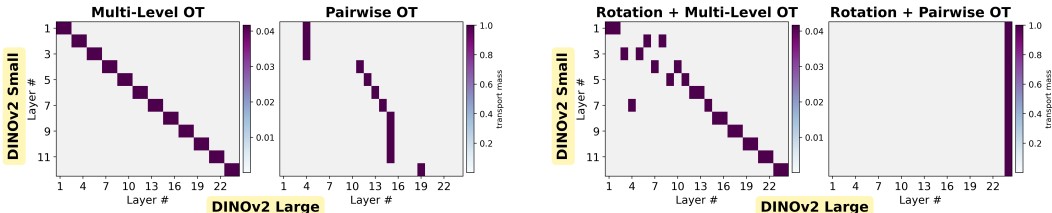

Figure D.2: **Transport plans for vision model alignment.** DINOv2 Small ↔ DINOv2 Large (a) without rotation (MOT) and (b) with rotation augmentation (MOT+R).

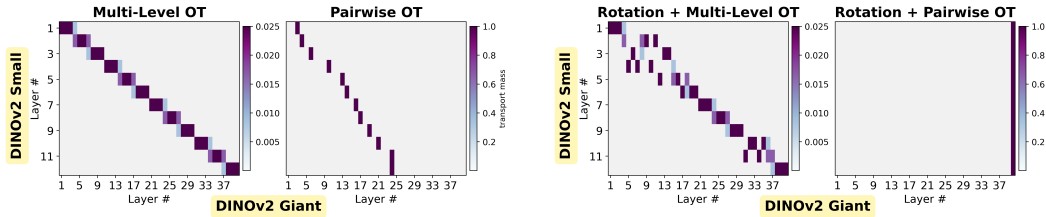

Figure D.3: **Transport plans for vision model alignment.** DINOv2 Small ↔ DINOv2 Giant (a) without rotation (MOT) and (b) with rotation augmentation (MOT+R).

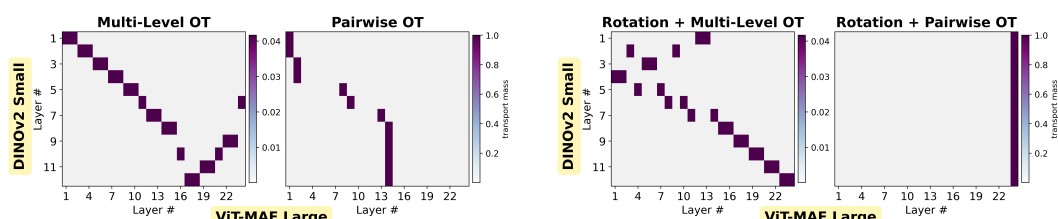

Figure D.4: **Transport plans for vision model alignment.** DINOv2 Small ↔ ViT-MAE Large (a) without rotation (MOT) and (b) with rotation augmentation (MOT+R).

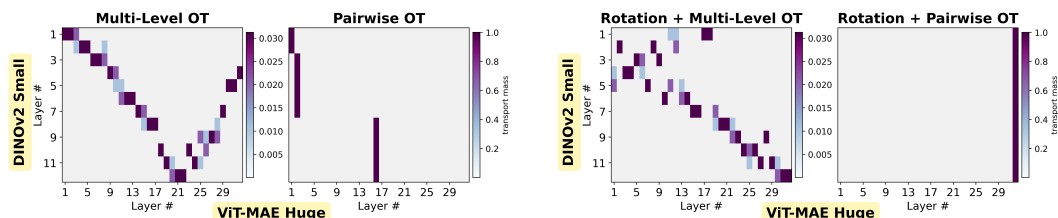

Figure D.5: **Transport plans for vision model alignment.** DINOv2 Small ↔ ViT-MAE Huge (a) without rotation (MOT) and (b) with rotation augmentation (MOT+R).

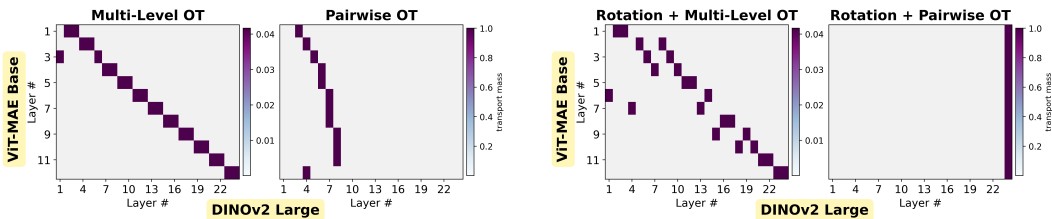

Figure D.6: **Transport plans for vision model alignment.** ViT-MAE Base ↔ DINOv2 Large (a) without rotation (MOT) and (b) with rotation augmentation (MOT+R).

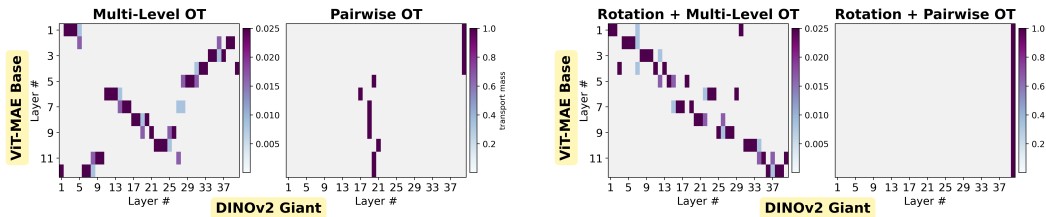

Figure D.7: **Transport plans for vision model alignment.** ViT-MAE Base ↔ DINOv2 Giant (a) without rotation (MOT) and (b) with rotation augmentation (MOT+R).

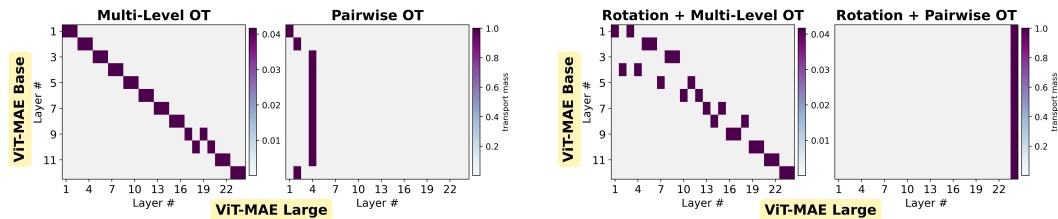

Figure D.8: **Transport plans for vision model alignment.** ViT-MAE Base ↔ ViT-MAE Large (a) without rotation (MOT) and (b) with rotation augmentation (MOT+R).

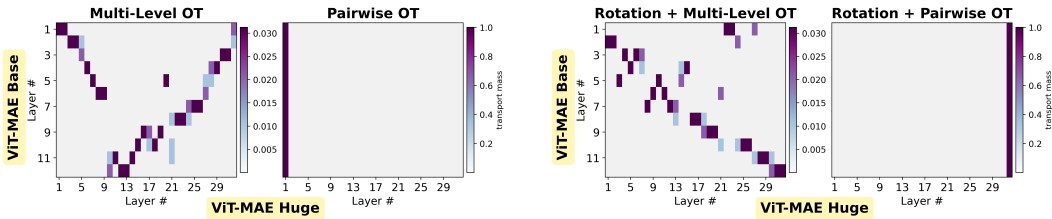

Figure D.9: **Transport plans for vision model alignment.** ViT-MAE Base ↔ ViT-MAE Huge (a) without rotation (MOT) and (b) with rotation augmentation (MOT+R).

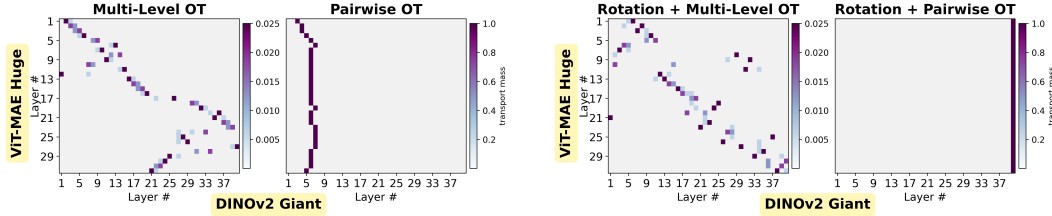

Figure D.10: **Transport plans for vision model alignment.** ViT-MAE Huge ↔ DINOv2 Giant (a) without rotation (MOT) and (b) with rotation augmentation (MOT+R).

# E REPRESENTATION SIMILARITY BETWEEN VISION MODELS AND VISION CORTEX

**Experimental Setup**. We assess representational similarity between human visual cortex and vision transformers by comparing fMRI responses with model activations elicited by the same set of stimuli. Specifically, we use responses from the Natural Scenes Dataset ((Allen et al., 2022)), focusing on 1,000 shared images viewed by participants in the fMRI experiment. The same images are presented to pretrained vision transformers, and layer-wise representations are extracted by averaging patch embeddings across each input. Following the approach used in the cortex–cortex analysis, we treat distinct visual areas (V1–V4) as "layers" and individual voxels as "neurons." We then compute both MOT and its rotation-augmented variant (MOT+R) to align cortical responses with model representations. This design enables a direct comparison of hierarchical organization across biological and artificial systems under matched visual input.

| Model 1 | Model 2 | MOT Metric | Pairwise Best OT | MOT + R | Pairwise Best + R |
|---------|---------|-----------|-----------------|---------|------------------|
| Subject | DINOv2 Base | 0.092 | 0.084 | **0.145** | 0.094 |
| Subject | DINOv2 Giant | 0.090 | 0.071 | **0.163** | 0.110 |
| Subject | ViT-MAE Base | 0.127 | 0.114 | **0.151** | 0.121 |
| Subject | ViT-MAE Huge | 0.072 | 0.099 | **0.154** | 0.122 |

Table E.1: Results on MOT metric vs. baselines (test split).

**Results**. Table E.1 reports reconstruction scores for MOT, MOT+R, and corresponding baselines averaged across the four subjects. Among all methods, MOT+R achieves the highest reconstruction score, indicating that incorporating rotation is critical for capturing shared structure between cortical and model representations. Subject-specific results (Tables E.2 and E.3) further confirm this trend, with MOT+R consistently outperforming both vanilla MOT and pairwise baselines. Transport plans visualized in Figures E.1– E.16 reveal partial but inconsistent layer-wise correspondences: in some cases, early cortical regions align with early model layers and higher regions map to deeper layers, while in others the mappings appear noisier.

| Model 1 | Model 2 | MOT Metric | Random (Perm-P) | Single-Best OT | Pairwise Best OT |
|---------|---------|-----------|----------------|---------------|-----------------|
| Subject A | DINOv2 Giant | 0.085 | 0.087 | 0.046 | 0.063 |
| Subject A | DINOv2 Base | 0.081 | 0.072 | 0.075 | 0.080 |
| Subject A | ViT-MAE Huge | 0.065 | 0.057 | 0.063 | 0.105 |
| Subject A | ViT-MAE Base | 0.141 | 0.131 | 0.123 | 0.124 |
| Subject B | DINOv2 Giant | 0.087 | 0.089 | 0.054 | 0.065 |
| Subject B | DINOv2 Base | 0.088 | 0.080 | 0.075 | 0.082 |
| Subject B | ViT-MAE Huge | 0.077 | 0.072 | 0.073 | 0.093 |
| Subject B | ViT-MAE Base | 0.115 | 0.109 | 0.101 | 0.104 |
| Subject C | DINOv2 Giant | 0.115 | 0.124 | 0.072 | 0.092 |
| Subject C | DINOv2 Base | 0.115 | 0.108 | 0.095 | 0.100 |
| Subject C | ViT-MAE Huge | 0.092 | 0.088 | 0.082 | 0.114 |
| Subject C | ViT-MAE Base | 0.137 | 0.134 | 0.121 | 0.124 |
| Subject D | DINOv2 Giant | 0.073 | 0.082 | 0.049 | 0.063 |
| Subject D | DINOv2 Base | 0.083 | 0.078 | 0.072 | 0.072 |
| Subject D | ViT-MAE Huge | 0.054 | 0.049 | 0.046 | 0.086 |
| Subject D | ViT-MAE Base | 0.114 | 0.111 | 0.101 | 0.106 |

Table E.2: Results on MOT metric vs. baselines (test split).

| Model 1 | Model 2 | MOT + R | Single-Best + R | Pairwise Best + R |
|---------|---------|---------|-----------------|-------------------|
| Subject A | DINOv2 Giant | 0.161 | 0.140 | 0.099 |
| Subject A | DINOv2 Base | 0.146 | 0.140 | 0.083 |
| Subject A | ViT-MAE Huge | 0.165 | 0.151 | 0.127 |
| Subject A | ViT-MAE Base | 0.157 | 0.151 | 0.124 |
| Subject B | DINOv2 Giant | 0.147 | 0.132 | 0.096 |
| Subject B | DINOv2 Base | 0.127 | 0.120 | 0.084 |
| Subject B | ViT-MAE Huge | 0.125 | 0.109 | 0.104 |
| Subject B | ViT-MAE Base | 0.128 | 0.118 | 0.101 |
| Subject C | DINOv2 Giant | 0.205 | 0.183 | 0.147 |
| Subject C | DINOv2 Base | 0.185 | 0.181 | 0.128 |
| Subject C | ViT-MAE Huge | 0.189 | 0.179 | 0.151 |
| Subject C | ViT-MAE Base | 0.186 | 0.180 | 0.152 |
| Subject D | DINOv2 Giant | 0.139 | 0.123 | 0.099 |
| Subject D | DINOv2 Base | 0.121 | 0.114 | 0.081 |
| Subject D | ViT-MAE Huge | 0.135 | 0.122 | 0.107 |
| Subject D | ViT-MAE Base | 0.134 | 0.129 | 0.109 |

Table E.3: Results on rotational MOT metric vs. baselines (test split).

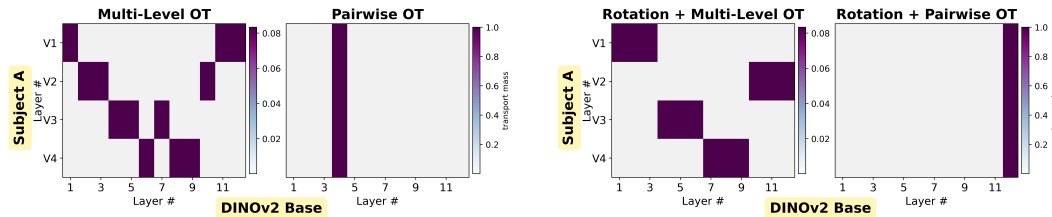

Figure E.1: Subject A ↔ DINOv2 Base (a) without rotation (MOT) and (b) with rotation augmentation (MOT+R).

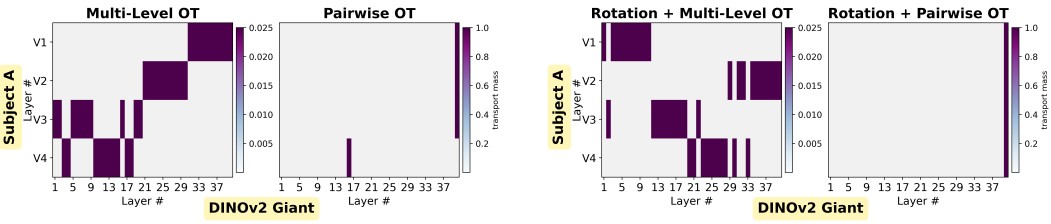

Figure E.2: Subject A ↔ DINOv2 Giant (a) without rotation (MOT) and (b) with rotation augmentation (MOT+R).

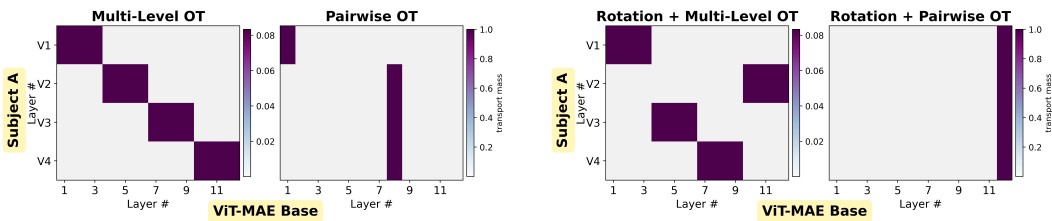

Figure E.3: Subject A ↔ ViT-MAE Base (a) without rotation (MOT) and (b) with rotation augmentation (MOT+R).

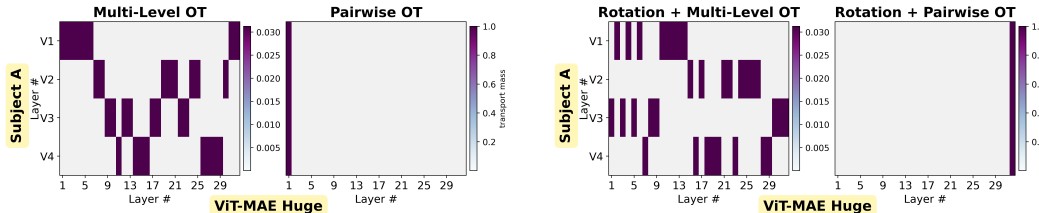

Figure E.4: Subject A ↔ ViT-MAE Huge (a) without rotation (MOT) and (b) with rotation augmentation (MOT+R).

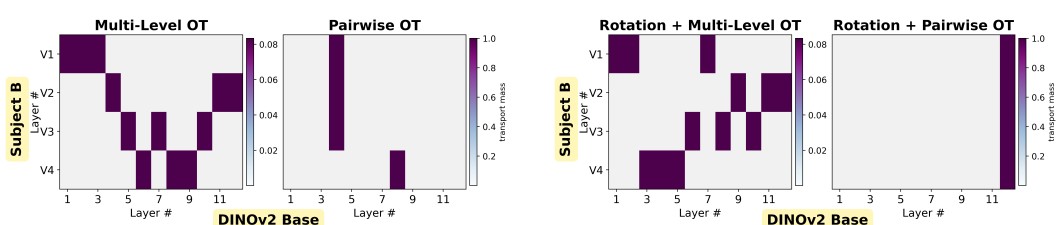

Figure E.5: Subject B ↔ DINOv2 Base (a) without rotation (MOT) and (b) with rotation augmentation (MOT+R).

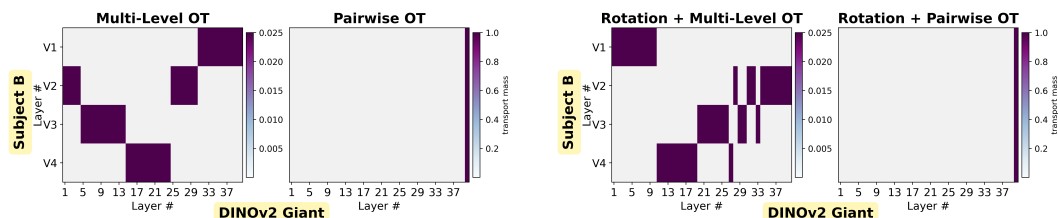

Figure E.6: Subject B ↔ DINOv2 Giant (a) without rotation (MOT) and (b) with rotation augmentation (MOT+R).

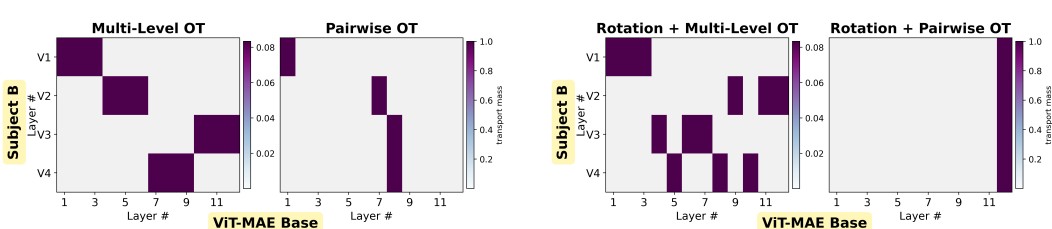

Figure E.7: Subject B ↔ ViT-MAE Base (a) without rotation (MOT) and (b) with rotation augmentation (MOT+R).

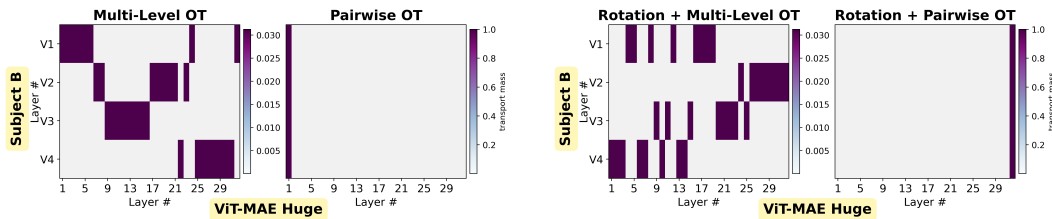

Figure E.8: Subject B ↔ ViT-MAE Huge (a) without rotation (MOT) and (b) with rotation augmentation (MOT+R).

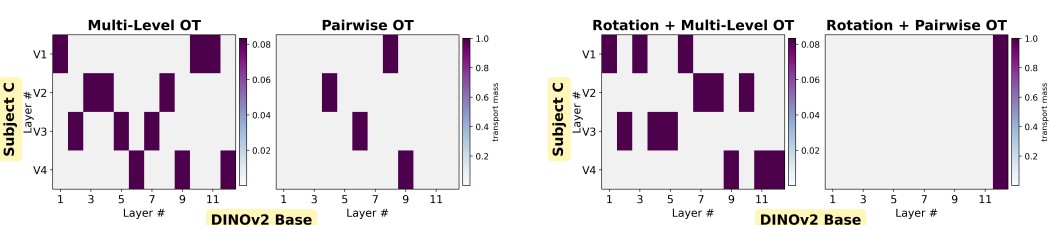

Figure E.9: Subject C ↔ DINOv2 Base (a) without rotation (MOT) and (b) with rotation augmentation (MOT+R).

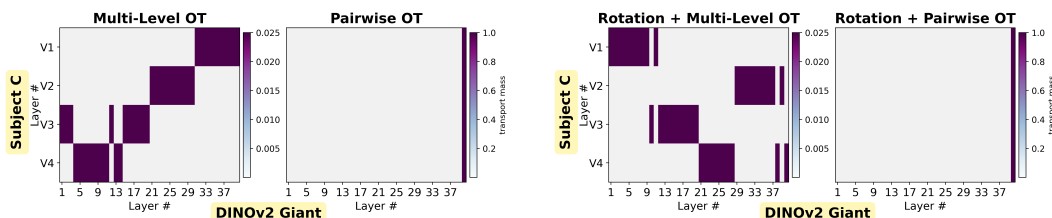

Figure E.10: Subject C ↔ DINOv2 Giant (a) without rotation (MOT) and (b) with rotation augmentation (MOT+R).

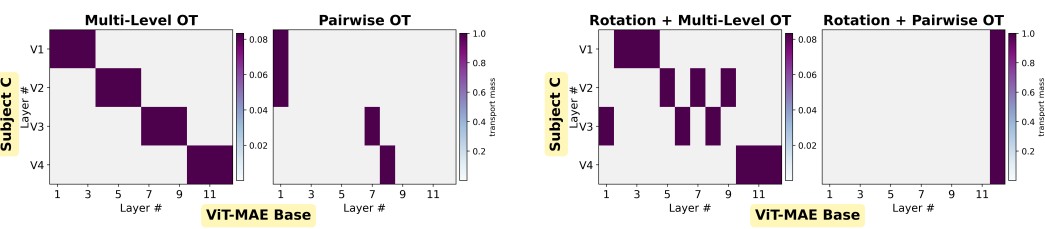

Figure E.11: Subject C ↔ ViT-MAE Base (a) without rotation (MOT) and (b) with rotation augmentation (MOT+R).

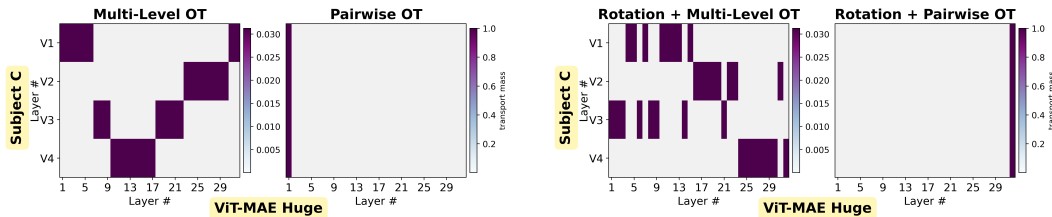

Figure E.12: Subject C ↔ ViT-MAE Huge (a) without rotation (MOT) and (b) with rotation augmentation (MOT+R).

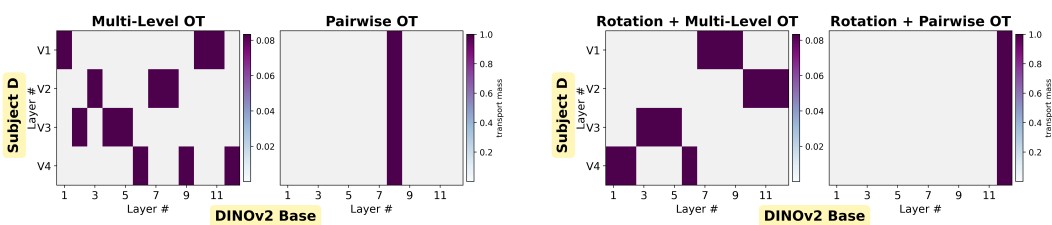

Figure E.13: Subject D ↔ DINOv2 Base (a) without rotation (MOT) and (b) with rotation augmentation (MOT+R).

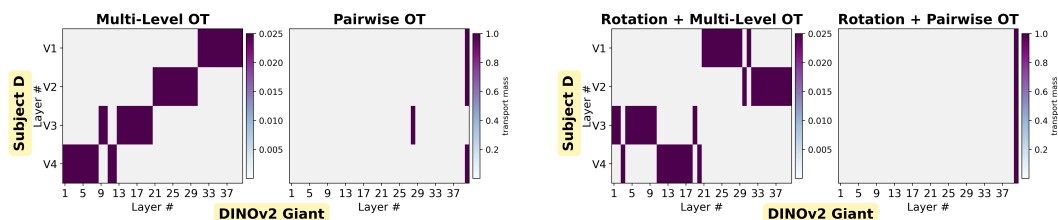

Figure E.14: Subject D ↔ DINOv2 Giant (a) without rotation (MOT) and (b) with rotation augmentation (MOT+R).

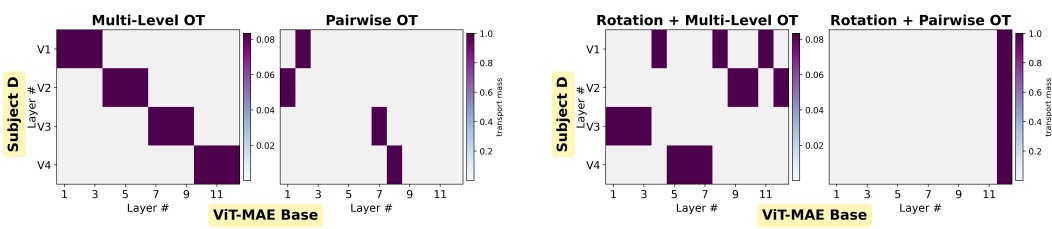

Figure E.15: Subject D ↔ ViT-MAE Base (a) without rotation (MOT) and (b) with rotation augmentation (MOT+R).

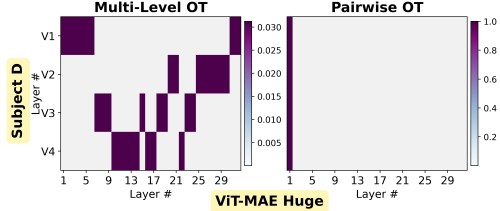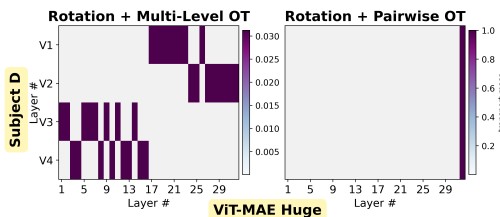

Figure E.16: Subject D ↔ ViT-MAE Huge (a) without rotation (MOT) and (b) with rotation augmentation (MOT+R).

# F  GLOBAL ALIGNMENT

In this section, we report reconstruction scores on a held-out validation dataset for the vision-cortex alignment study under a global alignment formulation. In this setting, all layers from both networks are flattened so that all neurons are treated as belonging to a single layer. We compare the resulting reconstruction scores to those obtained with our MOT reconstruction scores.

Global alignment scores are computed using two alignment metrics: optimal transport and linear predictivity. Reconstruction performance is then evaluated on held-out validation data. The results for this experiment are presented in Table F.1.

| Model 1 | Model 2 | Global OT | Global Linear Predictivity | MOT |
|---------|---------|-----------|----------------------------|-----|
| Subject A | Subject B | 0.184 | 0.305 | 0.244 |
| Subject A | Subject C | 0.177 | 0.295 | 0.199 |
| Subject A | Subject D | 0.168 | 0.289 | 0.198 |
| Subject B | Subject C | 0.183 | 0.306 | 0.212 |
| Subject C | Subject D | 0.158 | 0.306 | 0.197 |
| Subject B | Subject D | 0.163 | 0.297 | 0.201 |

Table F.1: Comparison of global OT, Linear Predictivity, and MOT, evaluated by reconstruction correlation on held-out fMRI responses.

# G  QUANTIFYING HIERARCHY

Given a layer-to-layer transport plan $P \in \mathbb{R}^{L \times M}$, we define a scalar "hierarchy" score that quantifies how strongly the transport mass is concentrated along a diagonal correspondence between source and target layers.

We first embed the layer indices into the unit interval via

$$r_\ell = \frac{\ell - 1}{L - 1}, \qquad c_m = \frac{m - 1}{M - 1}, \qquad \ell = 1, \ldots, L, \ m = 1, \ldots, M,$$

so that the ideal diagonal correspondence between layers lies along the line $r = c$. For each entry $(\ell, m)$ we define its normalized distance to this diagonal as

$$d_{\ell m} = \left| r_\ell - c_m \right| \in [0, 1].$$

We interpret the magnitudes of the transport plan as a discrete mass distribution,

$$s_{\ell m} = |P_{\ell m}|, \qquad w_{\ell m} = \frac{s_{\ell m}}{\sum_{\ell', m'} s_{\ell' m'}},$$

and compute the average distance of this mass from the diagonal,

$$D(P) = \sum_{\ell=1}^{L} \sum_{m=1}^{M} w_{\ell m} \, d_{\ell m}.$$

Finally, we define the hierarchy score as

$$H(P) = 1 - D(P).$$

By construction $H(P) \in [0, 1]$ and it depends only on the relative placement of transport mass with respect to the diagonal. Table G.1 reports $H(P)$ for each transport plan as a summary measure of its hierarchy.

| Model 1 | Model 2 | MOT | Pairwise OT | MOT+R | Pairwise Best + R |
|---|---|---|---|---|---|
| DINOv2 Small | ViT-MAE Large | **0.90** | 0.84 | 0.89 | 0.50 |
| ViT-MAE Base | ViT-MAE Large | **0.97** | 0.61 | 0.93 | 0.50 |
| DINOv2 Small | ViT-MAE Base | **1.00** | 0.88 | 0.85 | 0.50 |
| ViT-MAE Base | DINOv2 Large | **0.96** | 0.70 | 0.90 | 0.50 |
| DINOv2 Small | DINOv2 Large | **0.98** | 0.88 | 0.94 | 0.50 |
| ViT-MAE Huge | DINOv2 Giant | 0.86 | 0.62 | **0.86** | 0.50 |
| ViT-MAE Base | DINOv2 Giant | 0.60 | 0.55 | **0.91** | 0.50 |
| DINOv2 Small | DINOv2 Giant | **0.97** | 0.83 | 0.95 | 0.50 |
| ViT-MAE Base | ViT-MAE Huge | 0.68 | 0.50 | **0.83** | 0.50 |
| DINOv2 Small | ViT-MAE Huge | 0.82 | 0.72 | **0.88** | 0.50 |
| Llama-3.2 1B | Llama-3.2 3B | **0.96** | 0.93 | | |
| Qwen-2.5 0.5B | Llama-3.2 3B | **0.86** | 0.63 | | |
| Llama-3.2 3B | Qwen-2.5 3B | **0.83** | 0.80 | | |
| Llama-3.2 1B | Qwen-2.5 3B | **0.85** | 0.59 | | |
| Qwen-2.5 0.5B | Llama-3.2 1B | **0.86** | 0.67 | | |
| Qwen-2.5 0.5B | Qwen-2.5 3B | **0.93** | 0.60 | | |
| Subject B | Subject D | **1.00** | 0.93 | | |
| Subject C | Subject D | **1.00** | 0.98 | | |
| Subject B | Subject C | **1.00** | 0.93 | | |
| Subject A | Subject B | **1.00** | 0.98 | | |
| Subject A | Subject C | **1.00** | 0.98 | | |
| Subject A | Subject D | **1.00** | 0.98 | | |

Table G.1: MOT and pairwise (rotation where applicable) hierarchy scores for vision, language, and fMRI alignment.

We observe that, across all transport plans, the hierarchy metric is highest for plans generated by MOT (or MOT+R), compared with those produced by pairwise best (or pairwise rotational) methods. This indicates that MOT and MOT+R based approaches uncover hierarchical correspondences that pairwise methods don't reveal.

## H    TRANSPORT PLANS FOR LINEAR PREDICTIVITY BASED MAPPING PAIRS

| Model 1 | Model 2 | Pairwise LP Reconstruction score | MOT Metric |
|---|---|---|---|
| Llama-3.2 1B | Llama-3.2 3B | 0.737 | 0.558 |
| Qwen-2.5 0.5B | Llama-3.2 3B | 0.694 | 0.531 |
| Llama-3.2 3B | Qwen-2.5 3B | 0.784 | 0.383 |
| Llama-3.2 1B | Qwen-2.5 3B | 0.841 | 0.432 |
| Qwen-2.5 0.5B | Llama-3.2 1B | 0.848 | 0.522 |
| Qwen-2.5 0.5B | Qwen-2.5 3B | 0.848 | 0.510 |
| Subject B | Subject D | 0.210 | 0.201 |
| Subject C | Subject D | 0.230 | 0.197 |
| Subject B | Subject C | 0.199 | 0.212 |
| Subject A | Subject B | 0.238 | 0.244 |
| Subject A | Subject C | 0.179 | 0.199 |
| Subject A | Subject D | 0.206 | 0.198 |

Table H.1: **Pairwise Linear Predictivity and MOT reconstruction scores on test splits**. Top: LLM model-pair alignment. Bottom: fMRI subject-pair alignment.

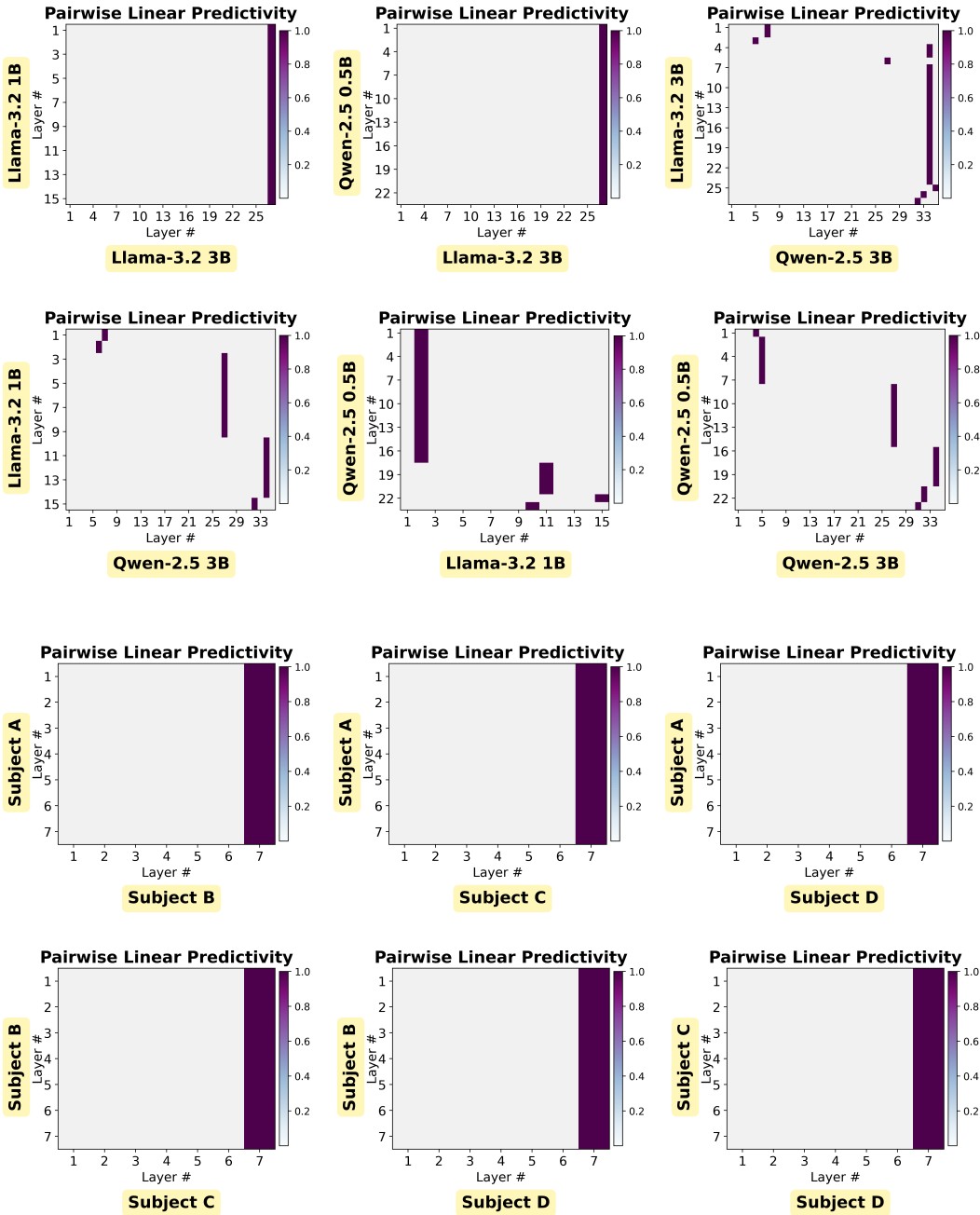

Figure H.1: **Transport plans for linear predictivity-based mappings in LLM and fMRI spaces.** Top two rows: pairwise mappings learned under linear predictivity constraints for all LLM pairs: (a) Llama-3.2 1B $\leftrightarrow$ Llama-3.2 3B, (b) Qwen-2.5 0.5B $\leftrightarrow$ Llama-3.2 3B, (c) Llama-3.2 3B $\leftrightarrow$ Qwen-2.5 3B, (d) Llama-3.2 1B $\leftrightarrow$ Qwen-2.5 3B, (e) Qwen-2.5 0.5B $\leftrightarrow$ Llama-3.2 1B, and (f) Qwen-2.5 0.5B $\leftrightarrow$ Qwen-2.5 3B. Bottom two rows: pairwise mappings learned under linear predictivity constraints for all fMRI subject pairs: (g) Subject A $\leftrightarrow$ Subject B, (h) Subject A $\leftrightarrow$ Subject C, (i) Subject A $\leftrightarrow$ Subject D, (j) Subject B $\leftrightarrow$ Subject C, (k) Subject B $\leftrightarrow$ Subject D, and (l) Subject C $\leftrightarrow$ Subject D. In both the LLM and fMRI settings, linear predictivity-based pairwise mappings do not recover structured layer-wise correspondences, highlighting the importance of MOT for learning robust and generalizable mappings across architectures, model scales, and subjects.

# I TRANSPORT PLANS FOR RSA-BASED MAPPING PAIRS

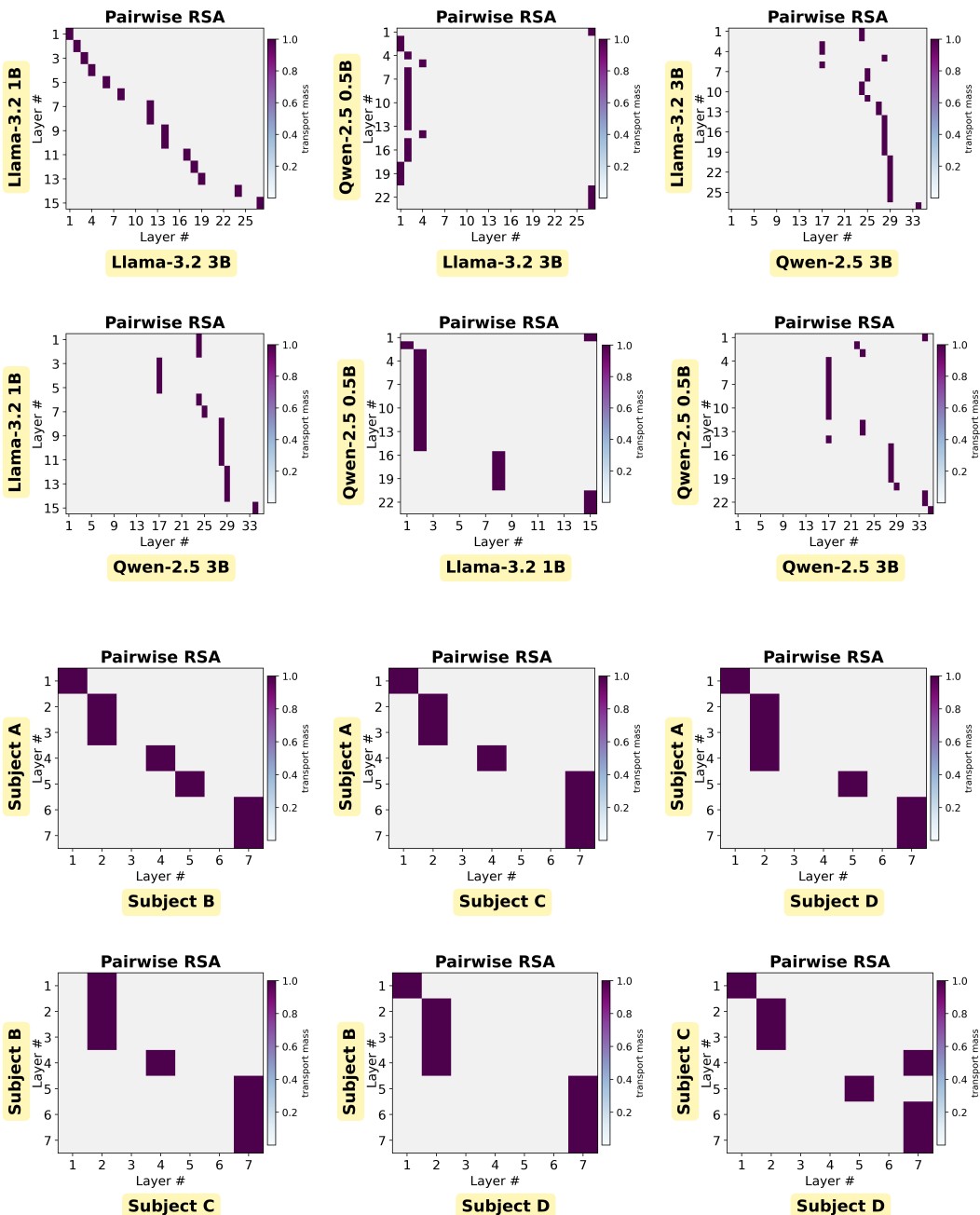

Figure I.1: **Transport plans for RSA-based mappings in LLM and fMRI spaces.** Top two rows: pairwise mappings learned under RSA-based constraints for all LLM pairs: (a) Llama-3.2 1B ↔ Llama-3.2 3B, (b) Qwen-2.5 0.5B ↔ Llama-3.2 3B, (c) Llama-3.2 3B ↔ Qwen-2.5 3B, (d) Llama-3.2 1B ↔ Qwen-2.5 3B, (e) Qwen-2.5 0.5B ↔ Llama-3.2 1B, and (f) Qwen-2.5 0.5B ↔ Qwen-2.5 3B. Bottom two rows: pairwise mappings learned under RSA-based constraints for all fMRI subject pairs: (g) Subject A ↔ Subject B, (h) Subject A ↔ Subject C, (i) Subject A ↔ Subject D, (j) Subject B ↔ Subject C, (k) Subject B ↔ Subject D, and (l) Subject C ↔ Subject D. In both the LLM and fMRI settings, RSA-based pairwise mappings do not recover structured layer-wise correspondences, highlighting the importance of MOT for learning robust and generalizable mappings across architectures, model scales, and subjects.

## J    ROBUSTNESS OF METRICS TO SUB-SAMPLING

We consider a pair of LLMs (Qwen-2.5 0.5B and Llama-3.2 1B) and systematically sub-sample neurons from the larger model (Llama3.2 1B) to study how this affects reconstruction of the smaller model's activations. For each fraction of neurons kept, we randomly select that fraction of neurons from Llama-3.2 1B, re-fit the transport plans between the two models, and evaluate the reconstruction score on held-out test data (a 20% split, as before), repeating this procedure over five different seeds. Figure J.1 shows the resulting reconstruction scores for MOT and MOT+R (with their corresponding pairwise baselines).

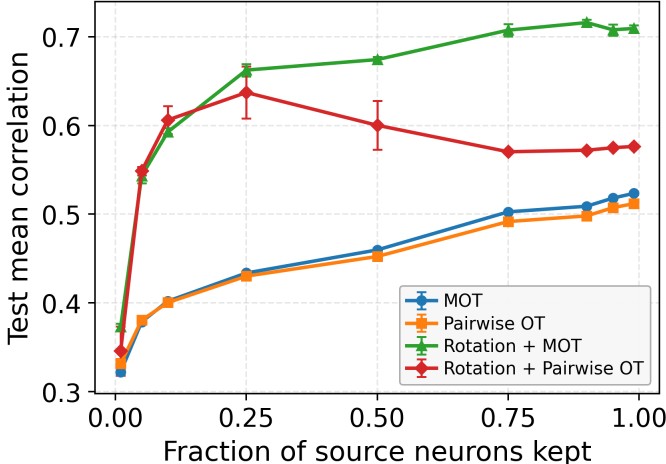

Figure J.1: **Neuron sub-sampling in LLM alignment.** Reconstruction performance when aligning Qwen-2.5 0.5B to Llama-3.2 1B as a function of the fraction of Llama neurons kept. The curves show reconstruction scores for MOT and MOT+R (with their corresponding pairwise baselines), averaged over five different seeds.

We observe that reconstruction scores for both MOT/MOT+R and the corresponding baselines increase with the fraction of neurons kept. Importantly, sub-sampling does not cause MOT to collapse; its performance degrades smoothly. For MOT+R, keeping only 10% of neurons yields a reconstruction score of nearly 0.5, indicating substantial robustness to aggressive sub-sampling. In the upper range of fraction of neurons kept, MOT+R achieves substantially higher reconstruction scores than both its rotation-aware baseline and the corresponding pairwise OT and MOT scores. These results suggest that neuron sub-sampling is a practical way to trade off computation and accuracy, and that the metrics remain informative even under significant sub-sampling.

## K    USE OF LARGE LANGUAGE MODELS

LLMs were used in this work to assist with writing tasks, specifically for ensuring grammatical correctness and enhancing the clarity of the text.

