# OpenReview forum: "Representational Alignment Across Model Layers and Brain Regions with Multi-Level Optimal Transport"
_ICLR.cc/2026/Conference — ICLR 2026 Poster_

### Official Review · Reviewer_8PNK · 2025-10-31

**Soundness:** 4
**Presentation:** 4
**Contribution:** 3
**Rating:** 8
**Confidence:** 2

**Summary:**

This paper introduces a way to compare what different neural networks (and even brains) represent across their layers, all at once, rather than matching layers and/or neurons one by one. Using a hierarchical optimal transport approach, it softly links each layer in one system to possibly several layers in another, producing a single overall alignment score and naturally handling models with different depths. Across vision models, language models, and brain imaging, the method matches or beats standard layer-by-layer matching and reveals clean, intuitive hierarchies, where early layers align with early layers and deeper with deeper, without hand crafting those rules.

**Strengths:**

- Original extension of the optimal transport framework to entire hierarchies, integrating many-to-many soft matching across layers and neurons in a unified manner; the method can handle models of different depths/sizes without hand-crafted layer pairing.
- Clearly written with conceptual claraity, and well explained.
- Empirical evaluation spans three relevant, diverse application domains.
- Candid discussion of limitations.
- The problem is important yet not well understood, so the potential impact is high.

**Weaknesses:**

As the limitations section notes, hierarchical OT is computationally costly, which is likely to limit the method's scalability. The somewhat narrow experimental scope leaves it unclear whether the method can effectively generalize to long-context settings, multimodal models, or tasks beyond vision and language.

Some typos remain  (e.g., 'and and' in line 086).

**Questions:**

Points that require further attention: (i) Sensitivity to the choice of hyperparameters. (ii) The correspondence between the observed alignment and behavior.

---

> ### Author Response · Authors · 2025-11-21
>
> We thank the reviewer for their positive feedback and below we address the questions raised by the reviewer.
>
> ***Computational Cost and Experimental Scope***
>
> `W1: “As the limitations section notes, hierarchical OT is computationally costly … tasks beyond vision and language.”`
>
>
> We agree that computational cost is an important practical limitation and appreciate the opportunity to clarify where it comes from and how it can be mitigated. The dominant cost in our method is inherited from the neuron-level OT solver, which we use as a building block. HOT does not introduce a qualitatively new solver; instead, it (i) reuses the same layer-wise OT at the neuron level and (ii) adds a second, much cheaper OT problem over layers. The only additional scaling factor relative to standard OT is that, to perform full model-to-model alignment, we solve OT for all layer pairs. This increases the number of OT problems (by a factor of L×M). The outer layer-level OT is low-dimensional and negligible in cost compared to neuron-level OT. Since the bottleneck is the underlying OT solver, HOT directly benefits from progress in scalable OT algorithms (e.g. entropic-regularized Sinkhorn iterations, mini-batch OT etc.). These methods can be dropped into our framework without any change to the hierarchical formulation, providing a clear path to more efficient implementations.
>
> Regarding the experimental scope, our goal in this paper was to demonstrate that HOT/HOT+R are general alignment tools by applying them across several qualitatively different settings rather than a single benchmark. Concretely, we study: (i) language-language alignment between distinct LLM families (Qwen2.5 and LLaMA3, up to 3B parameters), (ii) brain-brain alignment across human subjects, (iii) vision-vision alignment between different self-supervised ViT model families (DINOv2 and ViT-MAE at multiple scales), and (iv) vision-brain alignment (vision models to human visual cortex). These experiments already involve different modalities (language vs. vision), architectures (transformers and human cortex), and depth/width mismatches, all using full-size open-source models rather than toy networks. In the revised manuscript, we also add a new section extending our method to a three-level HOT and providing a proof of concept that HOT can recover checkpoint-level correspondences while aligning a pair of networks across their training trajectories; thereby further broadening the scope of our approach.
>
> More broadly, the formulation of HOT only assumes access to activation tensors over a set of stimuli; it is agnostic to the specific modality, task, or architectural details. In particular, long-context or multimodal models do not pose any conceptual obstacle: longer contexts primarily increase the number of time/sequence positions used to compute correlations (a cost that scales linearly in the number of samples), while the main OT complexity is governed by the number of units per layer and the number of layers which we already handle in the experiments above. Likewise, multimodal models or tasks beyond vision and language can be incorporated using the same formulation, including cases with differing depth and layer widths where HOT’s ability to handle depth and neuron mismatches is especially valuable. Exploring different settings is an exciting direction that we view as complementary to this work, although the current experiments already illustrate that HOT/HOT+R apply robustly across diverse systems.
>
>
> ***Choice of Hyper-parameters***
>
> `Q1: “Points that require further attention: (i) Sensitivity to the choice of hyperparameters…” `
>
> We thank the reviewer for raising this point and would like to clarify that our formulation of HOT and HOT+R is essentially hyperparameter-free at the metric level. Given a set of activations, the transport plans are obtained by solving well-defined OT problems with fixed constraints, and the resulting reconstruction scores are fully determined by the data and the choice of cost function (here, correlation-based), not by tunable hyperparameters. We therefore view the absence of dataset-specific hyperparameter tuning as a key advantage of HOT/HOT+R: our scores are directly comparable across settings without the extra layer of complexity and potential instability introduced by hyperparameter selection.

---

> > ### Author Response · Authors · 2025-11-21
> >
> > ***Correspondence between alignment and behaviour***
> >
> > `Q2: “Points that require further attention …  (ii) The correspondence between the observed alignment and behavior.”`
> >
> > We appreciate this observation. The question of which representational similarity measures capture functionally meaningful information, including whether representational alignment predicts behavioral similarity, is an active and important area of research that extends beyond the scope of the present work. Several recent studies have begun to investigate this systematically. For example, Bo et al. (2025) examine which metrics best predict downstream task performance, Ding et al. (2021) provide statistical procedures to assess whether similarity reflects shared computations rather than superficial correlations, and Harvey et al. (2024) offer a theoretical account linking geometric similarity to the information that can be linearly decoded from a representation.
> > Together, these works underscore that understanding how representational similarity relates to functional and behavioral relevance is a broader challenge for the field, one that applies to RSA, CKA, linear predictivity, soft matching, and other metrics. While HOT provides a principled and interpretable way to quantify representational alignment across processing stages of different networks, determining when and how this correspondence predicts behavior is an important direction for future work, which we now explicitly acknowledge in the revised Discussion.
> >
> > ---
> > ***References***
> >
> > * Bo, Yiqing, et al. "Evaluating representational similarity measures from the lens of functional correspondence." arXiv preprint arXiv:2411.14633 (2024).
> >
> > * Ding, Frances, Jean-Stanislas Denain, and Jacob Steinhardt. "Grounding representation similarity through statistical testing." *Advances in Neural Information Processing Systems* 34 (2021): 1556-1568.
> >
> > * Harvey, Sarah E., David Lipshutz, and Alex H. Williams. "What representational similarity measures imply about decodable information." arXiv preprint arXiv:2411.08197 (2024).
> >
> > ---
> >
> > We appreciate the reviewer’s encouraging comments and thank them again for their valuable feedback.

---

### Official Review · Reviewer_xsQG · 2025-11-01

**Soundness:** 4
**Presentation:** 4
**Contribution:** 4
**Rating:** 10
**Confidence:** 3

**Summary:**

The paper introduces Hierarchical Optimal Transport for representational alignment. Instead of matching each source layer to one best target layer in isolation, HOT matches neurons inside each candidate layer pair and, on top of that, a soft layer-to-layer coupling that must be globally consistent. This gives a single network-level alignment score, handles depth mismatches, and, the authors argue, reveals smooth hierarchical correspondences that greedy pairwise methods miss. The method also has a rotation-invariant variant (HOT+R) via alternating OT and Procrustes. Experiments span LLM to LLM, vision to vision, and NSD brain to brain. Vanilla HOT typically matches or beats pairwise in LLMs and brain, while HOT+R strongly improves on vision transformers, where rotation invariance is known to matter.

**Strengths:**

This work addresses an important challenge that a lot of representational similarity comparison studies currently face: how do we match the layers in a principled manner? This problem has been overlooked, but requires attention. Therefore, the significance of this work is high.

1. It is great that the authors introduce the algorithms for both rotation-sensitive and rotation-invariant similarity measures.

2. It is simple to extend the proposed method to incorporate more hierarchies. As noted in the discussion section, this work opens up a possibility to perform interesting matching experiments that were not possible before. One additional application with the third hierarchical level would be simply matching models (which would, at the lower level, match the layers, and then, at an even lower level, match neurons).

3. The authors provide reasonable baselines for proper comparisons

4. The results are remarkably interpretable and biologically useful. The high clarity in the presentation helps.

5. As far as I know, this work is original.

**Weaknesses:**

As already noted in the limitations section of the paper, HOT/HOT+R is computationally demanding. This can be alleviated by subsampling the neurons, particularly for HOT+R. I am guessing that subsampling the neurons for HOT would lead to a dramatically bad matching score, however. It would be nice if the authors could experimentally test how sensitive the matching (HOT/HOT+R) is to subsampling of neurons.

**Questions:**

Could the authors perform subsampling sensitivity analysis, as noted above?

---

> ### Author Response · Authors · 2025-11-21
>
> We thank the reviewer for their positive feedback and below we address the question raised by the reviewer.
>
> ***Subsampling neurons to reduce computational cost***
>
> `W1: “As already noted in the limitations section of the paper, HOT/HOT+R is computationally…”` \
> `Q1: “Could the authors perform subsampling sensitivity analysis, as noted above?”`
>
> We thank the reviewer for their suggestion and have added a subsampling sensitivity analysis. Specifically, we consider a pair of LLMs (Qwen2.5 0.5B and Llama3.2 1B) and systematically subsample neurons from the wider model (Llama3.2 1B), then measure how this affects the reconstruction of the smaller model’s activations. For each fraction of neurons kept (frac_neurons_kept), we randomly select that fraction of neurons from Llama3.2 1B, re-fit the transport plans between the two models, and evaluate the reconstruction score on held-out test data. The table below reports the resulting reconstruction scores.
>
> |   frac_neurons_kept |   HOT |   Pairwise Best OT |   HOT+R |   Pairwise Best + R |
> |---------------------------:|-----------------:|----------------------:|---------------------:|--------------------------:|
> |                     0.0100 |           0.3240 |                0.3188 |               0.3642 |                    0.3227 |
> |                     0.0500 |           0.3728 |                0.3710 |               0.5330 |                    0.5431 |
> |                     0.1000 |           0.4062 |                0.4037 |               0.6005 |                    0.6102 |
> |                     0.2500 |           0.4336 |                0.4276 |               0.6569 |                    0.6315 |
> |                     0.5000 |           0.4579 |                0.4504 |               0.6824 |                    0.7008 |
> |                     0.7500 |           0.5009 |                0.4921 |               0.6854 |                    0.5685 |
> |                     0.9000 |           0.5055 |                0.4963 |               0.7019 |                    0.5707 |
> |                     0.9500 |           0.5166 |                0.5052 |               0.7049 |                    0.5728 |
> |                     0.9900 |           0.5206 |                0.5101 |               0.6957 |                    0.5745 |
>
> We observe that reconstruction scores for both HOT/HOT+R and the corresponding baselines increase with the fraction of neurons kept. Importantly, subsampling does not cause HOT to collapse, its performance degrades smoothly. For HOT+R, we find that keeping only 10% of neurons already yields a reconstruction score of ~0.5, indicating substantial robustness to aggressive subsampling. In the upper range of frac_neurons_kept, HOT+R achieves substantially higher reconstruction scores than both its rotation-aware baseline and the corresponding HOT/pairwise HOT scores. These results suggest that neuron subsampling is a practical way to trade off computation and accuracy, and that HOT/HOT+R remain informative even under significant subsampling.
>
> ---
>
> We are grateful for the reviewer’s encouraging feedback and thank them once more for their valuable input.

---

> > ### Comment · Reviewer_xsQG · 2025-11-25
> >
> > Thank you for the response and for performing the analysis. I think the result can be better presented as a figure with error bars. Would it make sense to plot each reconstruction score as a function of the fraction of neurons kept, along with the error bars (randomness from sampling of neurons)?

---

> > > ### Author Response · Authors · 2025-12-03
> > >
> > > We thank the reviewer for their feedback. We have added a dedicated analysis in Supplementary Section J (Figure J.1), where we plot the reconstruction score as a function of the fraction of neurons kept, including error bars for different random sub-samples of neurons. This figure shows that reconstruction performance degrades smoothly with sub-sampling and that our metrics remain meaningful and robust even when only a subset of neurons is used.

---

### Official Review · Reviewer_1X33 · 2025-11-02

**Soundness:** 2
**Presentation:** 2
**Contribution:** 2
**Rating:** 4
**Confidence:** 2

**Summary:**

The paper proposes Hierarchical Optimal Transport (HOT), a unified framework for aligning representations across neural networks or between brains and models. Unlike standard pairwise layer-matching methods that treat each layer independently, HOT jointly infers globally consistent layer-to-layer and neuron-to-neuron transport plans under marginal constraints. This allows the transport “mass” to be distributed across multiple target layers while preserving global balance, yielding a single, symmetric alignment score for the entire hierarchy.
The authors apply HOT to compare representations across large language models, vision transformers, and human visual cortex recordings. They evaluate alignment using two criteria: (1) the average correlation between reconstructed and ground-truth responses, and (2) whether the inferred transport plans preserve hierarchical correspondences, such that layers at similar depths in the source and target systems align with each other

**Strengths:**

The paper addresses an important and timely problem in representational alignment—how to compare internal representations across networks and between brains and models when architectures differ in depth or structure. The proposed Hierarchical Optimal Transport (HOT) framework is an original and elegant reformulation of this problem, combining optimal transport with a hierarchical layer coupling that enforces global consistency. This formulation provides a principled way to compute both neuron-level and layer-level correspondences simultaneously, which is conceptually innovative compared to standard pairwise matching approaches.
The paper is generally well-written and situates the work in a broad and relevant context, bridging literature from both neuroscience and machine learning. The authors’ application of HOT across diverse domains—large language models, vision transformers, and human fMRI data—demonstrates methodological versatility and cross-domain relevance, which is a significant strength. The idea of producing a single, symmetric alignment score and a transport plan that naturally handles networks of different depths is both creative and potentially useful for future comparative studies of representation learning.

**Weaknesses:**

On Soundness:
The central claim of the paper is that Hierarchical Optimal Transport (HOT) provides a superior measure of representational alignment, with alignment “quality” quantified by the correlation between reconstructed and ground-truth neural responses. However, the empirical evidence does not convincingly support this claim. Across the three experimental domains, improvements are limited to the large language model comparisons, while results on human visual cortex and vision model alignments show marginal or no advantage. In fact, for vision models (Table 3), HOT underperforms the Pairwise Best OT baseline, and only the rotation-augmented version (HOT + R) shows improvement—suggesting that the observed gains stem primarily from the rotation component rather than the hierarchical formulation itself. Furthermore, critical baselines such as RSA, CKA, and linear predictivity are missing, and no measures of variance or statistical error are reported in Tables 1–3. As a result, the empirical support for the claimed superiority of HOT remains weak and incomplete.

On Presentation:
Presentation and clarity could be improved. A short introductory paragraph or visual summary explaining optimal transport theory in the context of representational alignment would also help orient readers unfamiliar with OT.
Figures and tables could be clearer and more polished. Axis ticks and labels are sometimes difficult to read. In Figure 2, the axis dimensions should be unified and limits matched across panels to make visual comparison meaningful; the overall aesthetics of the figure could also be improved. The tables listing model or subject-pair reconstruction scores might be better visualized as heatmaps, allowing all pairwise results to be compared at a glance. Additionally, in some heatmaps, it is unclear why certain axis labels are highlighted.
In the Results section for Experiment 3, the statement that the lack of a diagonal structure in transport plans is “consistent with” lower reconstruction scores is not well justified—this seems inconsistent with Experiment 2, where low reconstruction scores still coincided with clear hierarchical (diagonal) alignment patterns. Finally, the claims that HOT provides more generalizable and interpretable results are not clearly substantiated in the text: the paper does not explain what “generalizable” means in this context, and interpretability is equated only with hierarchical structure. However, whether layers at similar depths should align is itself an open empirical question, not an inherent criterion for interpretability.

On Contribution:
While the idea of combining hierarchical structure with optimal transport is interesting, the paper’s contribution is primarily incremental in practice. The empirical analyses do not convincingly demonstrate substantial improvements over existing OT-based methods, and the hierarchical formulation seems more interpretive than functionally necessary. Several important baselines (e.g., RSA, CKA, linear predictivity) and control analyses are missing, making it difficult to assess the true scope of the contribution. Moreover, claims about improved interpretability and generalizability remain qualitative and are not rigorously supported.

Expanded comment:
The empirical results do not convincingly support the central claims. The paper argues that layer-wise alignment is too rigid and proposes a hierarchical coupling (HOT) that distributes representational mass across layers. However, if meaningful correspondences truly span multiple layers, it is unclear why one should impose any hierarchical structure at all. A global all-to-all mapping—such as linear predictivity or optimal transport across all neurons, regardless of layer boundaries—could directly capture these distributed relationships without the additional constraint.

The authors do not justify why the hierarchical formulation is preferable or demonstrate that it yields empirical benefits beyond interpretability. In its current form, the hierarchical constraint seems imposed rather than validated, and the layer-level mass-balancing constraint may even bias the method toward producing smooth, diagonal correspondences that are visually appealing but potentially artifactual. A comparison with a fully unconstrained baseline (e.g., all-to-all linear mapping or global OT) would be critical to determine whether HOT provides genuine improvements in alignment accuracy or interpretability.
Additionally, incorporating statistical measures of uncertainty and comparing against established baselines such as RSA, CKA, and layer-wise linear predictivity would substantially strengthen the empirical case. Other recent OT-based methods such as Soft Matching Distance (Khosla & Williams, 2024) or model stitching (Bansal et al., 2021) are directly relevant and cited; including these would clarify whether the hierarchical layer coupling truly adds value over existing OT approaches.

Finally, there is ambiguity in the interpretation of results: in Experiment 3 (vision models), the authors interpret the lack of diagonal structure in transport maps as consistent with low reconstruction scores, but in Experiment 2 (visual cortex), low reconstruction scores still yielded diagonal correspondences. This inconsistency weakens the interpretation of what “hierarchical structure” signifies. A clearer, quantitative definition of hierarchy (e.g., correlation between layer indices or mutual information across depth) would improve interpretability. Moreover, the scalability of HOT remains untested in practice: although the paper notes a computational complexity of O(L2n3log⁡n), it would benefit from empirical runtime or memory comparisons with pairwise OT and linear predictivity to demonstrate its practical feasibility.

**Questions:**

1.	On methodological necessity:
 The paper argues that layer-wise alignment is too rigid and introduces a hierarchical coupling (HOT) that distributes representational mass across layers. However, if meaningful correspondences genuinely span multiple layers, why impose any hierarchical structure at all? Wouldn’t a global all-to-all alignment—such as linear predictivity or optimal transport across all neurons, irrespective of layer boundaries—capture these distributed relationships more directly? Could the authors clarify why the hierarchical formulation is necessary or advantageous compared to such an unconstrained alternative? At present, the hierarchical constraint seems imposed for interpretability rather than empirically justified. Additionally, might the layer-level mass-balancing constraint bias the model toward producing smooth, diagonal correspondences that appear hierarchical but could be artifactual? A comparison with a fully unconstrained baseline would help clarify whether HOT provides genuine advantages beyond aesthetic interpretability.


2.	On missing baselines and metrics:
 Why were standard representational alignment baselines such as RSA, CKA, Procrustes distance, and layer-wise linear predictivity not included in the evaluation? These are commonly used and directly comparable to the proposed method. Similarly, why were fully global baselines (e.g., all-to-all OT or global linear mapping) not tested, given that they would directly address whether HOT’s hierarchical constraint provides empirical benefits?


3.	On model and domain coverage:
 Why not also test HOT on convolutional neural networks (CNNs)? CNNs have known architectural inductive biases, and prior work (e.g., Yamins et al., 2014, PNAS) has shown that their layer activations align well with the ventral visual stream hierarchy. Including CNNs would clarify whether HOT can recover known brain-model hierarchical correspondences, not just those in transformers.
4.	On the biological data scope:
Why was the inferotemporal (IT) cortex excluded from the visual cortex analyses? IT is an established component of the ventral visual stream, and CNN layer–IT alignment using linear predictivity is well documented. Including IT data could reveal whether HOT captures higher-level cortical correspondences and provide a more complete test of its claims about hierarchical alignment.

5.	On representational choices:
For the transformer models, token activations were averaged to form layer-wise representations. Could the authors clarify what representational information this averaging preserves, and whether concatenating token embeddings or using other aggregation strategies changes the inferred alignment patterns?

---

> ### Author Response · Authors · 2025-11-21
>
> We thank the reviewer for the detailed review. We respond to each point below and provide ***multiple new experiments*** that further validate our claims.
>
> ***Hierarchical correspondence is emergent not imposed***
>
> `W4: “Expanded comment: The empirical results do not convincingly support the central claims. The paper argues that layer-wise alignment is too rigid and proposes a hierarchical coupling (HOT) … visually appealing but potentially artifactual.”`
>
> `Q1: “On methodological necessity: The paper argues that layer-wise alignment is too rigid and introduces a hierarchical coupling (HOT) that distributes representational … hierarchical but could be artifactual?”`
>
> We thank the reviewer for this thoughtful critique.
> First, we would like to correct an important misconception underlying the reviewer’s concern. Although we refer to our method as “Hierarchical OT,” the algorithm has no access to layer indices, layer ordering, or any notion of hierarchy. The optimization operates solely on representation matrices and a cost tensor; nothing in the formulation encodes which layers are “early,” “middle,” or “late,” nor does it include any smoothness, positional, or adjacency priors over layer indices. Therefore, HOT cannot be biased toward producing diagonal (hierarchy-preserving) couplings because it has no knowledge of what a diagonal even represents. The smooth, near-monotonic correspondences observed in Figures 2–4 and Appendix A–D are thus fully emergent from the global optimal transport objective, not imposed by any structural constraint.
> We also agree that, in principle, one could ignore layer boundaries and perform an all-to-all mapping across all neurons in the two systems (via linear predictivity or OT on flattened representations). However, we see three concrete reasons why this is not a preferable alternative to HOT.
>
> 1. **Computational complexity and scalability:**
>    Let each network have $L$ layers with $n$ units per layer (total $Ln$ units). HOT solves OT only at the layer level: inner OT between layers of size $n$ has cost $O(n^3 \log n)$, and we do this for $O(L^2)$ layer pairs, giving an overall $O(L^2 n^3 \log n)$ complexity (as stated in the paper). In contrast, a single all-to-all OT between the flattened populations of size $Ln$ is $O((Ln)^3 \log(Ln)) = O(L^3 n^3 \log(Ln))$. Thus, the all-to-all OT baseline is more expensive by a factor of $O(L)$, which is substantial for modern models (e.g., 24–80 layers). A similar issue arises for all-to-all linear mappings: fitting a linear predictor from $Ln$ units to $Ln$ units has a complexity of the order of $O(M(Ln)^2 + L^3 n^3)$ with $M$ stimuli. In contrast, the linear predictivity approach applied pairwise between all layer combinations would have complexity of the order of $O(ML^2 n^2 + L^2 n^3)$, again almost less by a factor of $O(L)$. In short, the all-to-all variants do not scale well to the network sizes we (and the community) care about.
>
> 2. **Ill-conditioning and over-flexibility of all-to-all mappings:**
>    The flattened approach suffers from two critical pathologies. First, it is too flexible: it can freely redistribute variance across all $Ln$ units, making the solution space enormous, underdetermined and thus prone to overfitting noise. Second, by deliberately ignoring that neurons are organized into sequential processing stages, the method fails to exploit computational structure that empirically exists in both biological and artificial neural networks. HOT is better conditioned precisely because it serves as a principled form of regularization that encodes well-established prior knowledge about how neural computation proceeds in stages. Consequently, while the transportation polytope is convex and in principle allows uniform spreading of mass, the optimal solution in practice is driven toward sparse, concentrated assignments.
>    To further justify this claim, we conducted the exact experiment the reviewer suggests to test whether the nested optimal transport structure inherent in OT genuinely improves performance or merely imposes artificial constraints. We flattened all layers into single global representations and applied standard optimal transport across all neurons simultaneously, ignoring layer boundaries entirely. We found that hierarchical OT performs better reconstruction relative to global OT; please refer to our response in “Global Alignment using OT and LP” for more details.
>
> 3. **No ability to localize correspondence across representational stages:**
>    Even if all-to-all linear predictivity or OT were computationally tractable, such mappings would no longer allow us to ask where in the hierarchy alignment occurs, e.g. early vs. late layers, or V1/V2 vs. V4, etc. nor to inspect which sets of neurons in one “stage” correspond to which in another. This type of approach would thus be far less informative about hierarchical correspondence.

---

> ### Author Response · Authors · 2025-11-21
>
> ***Performance of HOT and HOT+R for reconstruction***
>
> `W1: “On Soundness: The central claim of the paper is that Hierarchical Optimal Transport (HOT) provides a superior measure of representational alignment ... the hierarchical formulation itself.”`
>
> The hierarchical OT framework (with or without rotations, depending on the domain) yields alignment quality that is competitive with or better than standard pairwise OT baselines, while additionally providing a globally consistent alignment. Across the three domains we study, the empirical results support this claim:
>
> **(i) Language Model Alignment:**
> In all six LLM pairs, vanilla HOT achieves higher reconstruction correlation than other baselines (Table 1), including Pairwise Best OT. The gains are consistent and hold across both intra-family and cross-family comparisons. Thus, in the LLM regime, HOT clearly improves alignment quality under the reconstruction metric defined in the paper.
>
> **(ii) Visual Cortex Alignment:**
> For visual cortex alignment, HOT and Pairwise Best OT have indistinguishable reconstruction performance. Across all subject pairs, the absolute difference between HOT and Pairwise Best OT is smaller than the standard deviation over the 5 train/validation splits reported in Table C.1. Thus, HOT matches the reconstruction quality of the pairwise baseline rather than meaningfully underperforming it. More importantly, HOT generates transport plans which reliably align corresponding visual areas across individuals (e.g., V1–V1, V2–V2; Fig. C.1), which is expected; and the pairwise baseline fails to recover such region-to-region correspondences.
>
> **(iii) Vision Model Alignment:**
> We agree that, when evaluated in the rotation-sensitive setting, vanilla HOT does not consistently outperform Pairwise Best OT. However, for Vision Transformers the residual stream is known to be approximately rotation-invariant, so a rotation-invariant comparison is the more appropriate setting. In this regime, the hierarchical HOT+R variant strictly outperforms its rotation-aware pairwise baseline (Pairwise Best + R) for all model pairs in Table 3, sometimes by a large margin. If the improvements were due “only to rotation,” one would expect HOT+R and Pairwise Best + R to perform similarly, but empirically HOT+R is consistently better. This indicates that the combination of hierarchical coupling and rotation, not rotation alone is responsible for the gains.
>
> ***Global Alignment using OT and LP***
>
> `W4: “A comparison with a fully unconstrained baseline … interpretability.”`
>
> `Q1: “A comparison with a fully unconstrained baseline would help clarify whether HOT provides genuine advantages beyond aesthetic interpretability.”`
>
> `Q2: “Similarly, why were fully global baselines … empirical benefits?”`
>
> We thank the reviewer for suggesting a comparison with fully unconstrained baselines. In the revised version, we include new experiments on cortex-cortex alignment that directly address this point (Appendix Section F).
>
> We construct fully global baselines by flattening all layers in each subject into a single “layer,” treating all voxels as a single population, and then (i) Global OT: applying neuron-level OT between all source and target voxels at once (no hierarchical structure). (ii) Global linear mapping: fitting a single ridge regression from all source voxels to all target voxels. We evaluate these methods using the same reconstruction correlation on a held-out test set as for HOT. As reported in Table F.1, we find that: (i) Global OT yields lower reconstruction scores than HOT across subject pairs. This shows that the hierarchical constraint is not merely an aesthetic choice: when we remove it and allow mass to move freely across all regions, both alignment accuracy and interpretability degrade. (ii) Global linear mapping achieves higher reconstruction scores than HOT (and OT), which is expected because it is strictly more flexible: it can implement any affine transformations on the representational space. However, this flexibility comes at a cost: OT-based methods preserve the full structure of tuning functions by enforcing a transport plan that must account for all neuronal mass, ensuring that every unit in one system is matched to a soft-combination of units in the other, and metrics such as RSA or procrustes preserve representational geometry. A global linear map may arbitrarily mix, scale, and distort activations, so a high prediction score reflects only the existence of some shared linear subspace, not any preservation of tuning or geometry. We discuss the limitations of linear predictivity in more detail in our response to the comment on “pairwise comparisons using other metrics.”

---

> ### Author Response · Authors · 2025-11-21
>
> ***Pairwise comparisons using other metrics (RSA, LP)***
>
> `W1: “Furthermore, critical baselines such as RSA, CKA, and linear predictivity are missing … incomplete.” `
>
> `W3: “On Contribution: While the idea of combining hierarchical structure with optimal transport is interesting … contribution.”`
>
> `W4: “Additionally, incorporating statistical measures … adds value over existing OT approaches.”`
>
> `Q2: “On missing baselines and metrics … proposed method.”`
>
> We thank the reviewer for suggesting these additional baselines. Our response has two parts: (i) what transport maps RSA and layer-wise linear predictivity produce in our setups, and (ii) how their scores and limitations compare to HOT.
>
> **(i) Transport maps from pairwise RSA and linear predictivity**
>
> We perform pairwise layer-wise linear predictivity and RSA on the LLM alignment setup and the brain–brain alignment setup. We observe that neither the RSA metric nor the linear predictivity metric is able to reveal a hierarchical correspondence in either the LLM data or the brain data. In contrast, HOT does recover hierarchical correspondence in both the brain and LLM settings. The hierarchical correspondence is particularly expected in the brain data, because regions across individuals should align with each other when calculating the transport plan. The corresponding RSA and LP transport maps and scores are reported in Figure I.1, and Figure H.1 of the revised manuscript.
>
> **(ii) Reconstruction scores and downside of RSA/Linear Predictivity**
>
> Regarding the raw scores, RSA does not provide a reconstruction score because it is a measure of similarity. Linear predictivity does provide a reconstruction score, and we observe that the layer-wise linear predictivity score is better than the hierarchical optimal transport score (Table H.1). However, linear predictivity comes with its own issues because it is very flexible and this flexibility has some issues. In the next section we discuss various issues associated with RSA and linear predictivity and why it is not an appropriate metric to use all the time. Given the computational cost of running HOT and its baselines across many model pairs and large datasets, it was not feasible to perform multiple data splits for variance estimates in all settings; where it was feasible (cortex-cortex alignment), we do report mean scores and standard deviation over 5 splits in Table C.1.
>
> We also clarify why several commonly used baselines are not directly comparable to HOT in purpose or capability. RSA (and CKA) operate on similarity matrices rather than on the representations themselves. As a result, they abstract away the underlying feature space: all rotations, reflections, and permutations of units are treated as identical. This makes them useful for coarse geometric comparisons but unsuitable for prediction or for recovering neuron-level correspondences, and they cannot assess alignment of tuning functions. Linear predictivity (LP) sits at the opposite extreme. Because a linear map can absorb arbitrary affine transformations, including rotations, permutations, anisotropic scalings, and shear, LP is too flexible to preserve structure in the underlying geometry. Unlike OT (which enforces mass conservation) or Procrustes distance (which preserves shape), LP does not constrain how one representation is warped into another, making it unable to distinguish tuning-level structure or detect many qualitative differences that matter for representational similarity assessment. In addition, LP is asymmetric by construction (the “distance’’ from A→B differs from B→A), which makes it difficult to use as a metric of similarity between two systems. Recent empirical and theoretical work further highlights these limitations: Bo et al. show that LP correlates weakly with behavioral correspondence compared to geometry-respecting methods. Khosla & Williams, Bo et al. demonstrate that OT-based distances (including Soft Matching Distance) discriminate effectively between architectures such as CNNs and ViTs, whereas LP largely fails at this task precisely because it is too flexible. Avitan & Golan demonstrate that linear probes can fail to recover the generating representation even with very large datasets because their flexibility distorts underlying geometry. These findings illustrate that LP, RSA, and CKA answer different questions than HOT: they either abstract away geometry entirely (RSA/CKA) or allow arbitrary geometric distortions (LP).

---

> ### Author Response · Authors · 2025-11-21
>
> ***Edits to manuscript for clarity***
>
> `W2: “On Presentation: Presentation and clarity … help orient readers unfamiliar with OT.”`
>
> Thank you for the helpful suggestion. To better orient readers who may be unfamiliar with optimal transport (OT), we have added a dedicated subsection titled “Optimal Transport for Representational Alignment’’ in Appendix A (in red). This new text provides a concise introduction to OT in the specific context of comparing neural population codes. In particular, it explains how population responses can be viewed as probability distributions over tuning functions, how a generic dissimilarity cost between tuning curves defines a ground metric, and how the 2-Wasserstein (soft-matching) formulation computes the minimal “effort’’ required to transform one representational distribution into another. We also formally present the transportation polytope (which defines the transport plans) with its three key constraints: non-negativity, row marginals, and column marginals. Crucially, we emphasize key properties that make OT particularly well-suited for neural data: it accommodates unequal population sizes through soft assignments rather than requiring strict one-to-one matches, it preserves single-neuron tuning structure unlike rotation-invariant metrics such as RSA or CKA, and it yields explicit, interpretable neuron-to-neuron correspondences rather than reducing alignment to opaque summary scores.
>
> ***Hierarchical Correspondence: Correlation with Performance and Interpretability***
>
> `W2: “In the Results section for Experiment 3 … for interpretability.” `
>
> `W3: “Moreover, claims about improved interpretability and generalizability remain qualitative and are not rigorously supported.”`
>
> `W4: “Finally, there is ambiguity in the interpretation of results: … “hierarchical structure” signifies.” `
>
> We would like to thank the reviewer for pointing this out. We make no claims about a correlation between the reconstruction performance and the amount of hierarchical correspondence in the layer-wise transport plan. We have edited the manuscript and removed all statements that suggested a link between hierarchical/diagonal structure and reconstruction performance.
>
> What we mean by generalizable is that the reconstruction scores of the alignment algorithm on test data are higher than the corresponding baselines while also producing transport plans that are more sensible (for example, in the cortex-cortex alignment, HOT aligns corresponding regions across individuals).
>
> Regarding interpretability, we call our method interpretable because it generates a layer-wise and neuron-level transport plan, which allows us to see exactly what the optimization process is doing to align the representations showing which layers correspond across systems and, within each paired layer, which neurons are matched to one another. This level of detail is not available for other methods such as linear predictivity. Although it is not known that layers at similar depths “should” align in general or that this constitutes a ground truth, several influential studies, such as Kornblith et al’s CKA paper, explicitly use depth-wise alignment as a basis for metric selection. More importantly, in our cortex–cortex experiments we observe that HOT aligns the same visual regions across brains whereas baseline methods do not. This provides a concrete basis for interpretability as hierarchical progression is well-established across individuals, so correspondences between homologous regions is strongly expected. Because ground-truth mappings are rarely available in representational comparison, we adopt the same convention used in prior work introducing new tools, i.e., assessing whether a method recovers sensible depth-structured layer/region matches (Kornblith et al. (2019), Taylor et al. (2025), Thobani et al. (2025) ).

---

> ### Author Response · Authors · 2025-11-21
>
> ***Quantitative definition of hierarchy***
>
> `W4: “A clearer, quantitative definition of hierarchy (e.g., correlation between layer indices or mutual information across depth) would improve interpretability.”`
>
> We thank the reviewer for this suggestion. In the revised manuscript we add a new quantitative hierarchy measure in Supplementary Section G. Given a layer-to-layer transport plan $P$, we define a scalar hierarchy score $H(P)$ in $[0, 1]$ that measures how strongly the transport mass is concentrated along a diagonal correspondence between source and target layers. Intuitively, we embed source and target layer indices onto a common $[0, 1]$ scale, compute the average distance of the transport mass in $P$ from the ideal diagonal (which corresponds to perfect depth-wise alignment), and define $H(P)$ as one minus this average distance. Thus, higher $H(P)$ means more mass near a diagonal correspondence, while lower $H(P)$ indicates a more diffuse or off-diagonal plan.
>
> We report this hierarchy score for all transport plans in Table G.1 (Supplementary Section G). Across all settings, the hierarchy metric is consistently highest for transport plans generated by HOT (or HOT+R), compared to those produced by the pairwise baselines (including rotational pairwise). This quantitatively confirms that HOT/HOT+R uncover more hierarchical layer correspondences than the pairwise methods, in line with our qualitative visualizations.
>
> ***Model architecture coverage***
>
> `Q3: “On model and domain coverage … just those in transformers.”`
>
> Thank you for the suggestion. Our central contribution is methodological: introducing a principled framework for globally consistent hierarchical alignment. Our goal thus is to evaluate HOT in settings where we have well-established hierarchical expectations such as within vision models, within language models, and across cortical areas so that the quality of the recovered layer correspondences can be meaningfully interpreted. In these domains, hierarchical structure is either architecturally defined (model depth) or neuroanatomically documented (visual cortex hierarchy from V1 through high-level ventral visual stream), providing clear ground truth against which to validate whether HOT successfully recovers expected early-to-early and deep-to-deep correspondences.
>
> Brain-model comparisons, by contrast, present more ambiguous validation challenges. The reviewer cites Yamins et al. (2014) as evidence for known hierarchical correspondences between CNNs and primate ventral stream. However, subsequent work has revealed substantial complexity in these correspondences that complicates their interpretation as validation targets. For instance, Cadena et al. (2019) showed that the "optimal" layer for predicting any given brain area depends strongly on stimulus preprocessing choices such as input resolution, raising questions about whether these correspondences reflect genuine computational alignment or incidental feature-scale matching. Similarly, recent work by Soni et al. (2024) has shown that some representational similarity metrics fail to reveal hierarchical correspondence between CNNs and brains, further underscoring that brain-model hierarchical alignment is not a settled benchmark but rather an open empirical question whose answer depends on species, metric choice, and preprocessing decisions.
>
> Additionally, comparing convolutional layers introduces substantial technical complications because their representations are spatial tensors with built-in translation equivariance. Metrics that ignore this structure (e.g., by flattening) can treat very different spatial activation patterns as equivalent or fail to account for shifts in the receptive field. Recent work (Williams et al. 2021) has emphasized that proper comparison of convolutional layers must consider spatial alignment and channel-wise rotations, making the problem significantly more computationally expensive than comparing vector-valued hidden states.

---

> ### Author Response · Authors · 2025-11-21
>
> ***Extending brain area coverage***
>
> `Q4: “On the biological data scope … provide a more complete test of its claims about hierarchical alignment.” `
>
> We thank the reviewer for this suggestion. In the revised manuscript, we have expanded our visual cortex–visual cortex alignment experiments to include higher-level visual regions in addition to V1-V4. Specifically, we now also incorporate lateral, dorsal, and ventral visual cortex responses alongside V1-V4. This provides broader coverage of the ventral visual stream, including higher-level regions.
>
> Our main conclusions remain unchanged. The reconstruction score of HOT is very similar to that of the pairwise baseline. However, HOT is able to align similar regions across brains, whereas the pairwise method does not recover this region-to-region correspondence. We have updated all relevant figures and numbers in the main paper to reflect these extended analyses (Table 2, Figure 3, Table C.1, Figure C.1).
>
> We also repeat this expanded cortex-cortex experiment using a 5-fold train/test split. The mean and standard deviation of the reconstruction scores for HOT and the pairwise baseline are reported in Supplementary Section C, Table C.1. Although we observe that the mean score of HOT is very slightly lower compared to that of the pairwise method; the difference between the two methods is smaller than the standard deviation across splits, indicating that their reconstruction performance is effectively the same for our experiment setup.
>
> ***Choice of representations***
>
> `Q5: “On representational choices … the inferred alignment patterns?”`
>
> For the transformer models, we average token activations within each layer to obtain a single vector per stimulus. In language models, input sequence lengths vary across stimuli, so concatenating token embeddings would produce representations with different dimensionalities across inputs, which would break the alignment setup because the feature dimensionality would no longer be fixed across stimuli. Averaging provides a fixed-size representation per layer, aggregates information from all positions, and controls for sequence length.  Our representation choice follows standard practice in sentence-embedding work (e.g., Reimers & Gurevych, 2019), where mean-pooling over transformer tokens is widely used to produce stable, semantically meaningful sentence representations. For vision transformers, the input images are converted to patches before being fed to the ViT. Using all patch embeddings directly would again lead to extremely high-dimensional representations. Here we again average over all patch tokens to obtain a single layer-wise representation for each image. This choice mirrors the language-model setup, yields manageable dimensionality, and allows us to incorporate information from all spatial locations. This choice is also consistent with ViT’s original analysis: Dosovitskiy et al. (2021) report that a classifier using average pooling over patch embeddings performs similarly to the standard [CLS]-token classifier, suggesting that simple averaging preserves task-relevant information in the patch representations.
>
> ---
>
> Thank you again for the thoughtful and detailed feedback, it has helped us improve our work a lot. If our rebuttal adequately addresses your concerns, we **kindly ask you to consider raising your score**. We are happy to clarify any remaining questions you might have!

---

> ### Author Response · Authors · 2025-11-21
>
> ***References***
>
> * Kornblith, Simon, et al. "Similarity of neural network representations revisited." International conference on machine learning. PMLR, 2019.
>
> * Thobani, Imran, et al. "Model-brain comparison using inter-animal transforms." arXiv preprint arXiv:2510.02523 (2025).
>
> * Taylor, JohnMark, et al. “Framed RSA: honoring representational geometry and regional-mean response preferences.” Journal of Vision, [https://doi.org/10.1167/jov.25.9.2761](https://doi.org/10.1167/jov.25.9.2761).
>
> * Khosla, Meenakshi, and Alex H. Williams. "Soft Matching Distance: A metric on neural representations that captures single-neuron tuning." Proceedings of UniReps: the First Workshop on Unifying Representations in Neural Models. PMLR, 2024.
>
> * Avitan, Itamar, and Tal Golan. "Model-Behavior Alignment under Flexible Evaluation: When the Best-Fitting Model Isn't the Right One." arXiv preprint arXiv:2510.23321 (2025).
>
> * Bo, Yiqing, et al. "Evaluating representational similarity measures from the lens of functional correspondence." arXiv preprint arXiv:2411.14633 (2024).
>
> * Cadena, Santiago A., et al. “How well do deep neural networks trained on object recognition characterize the mouse visual system?” Real Neurons & Hidden Units: Future Directions at the Intersection of Neuroscience and Artificial Intelligence @ NeurIPS 2019, 2019. [https://openreview.net/forum?id=rkxcXmtUUS](https://openreview.net/forum?id=rkxcXmtUUS).
>
> * Soni, Ansh, et al. "Conclusions about neural network to brain alignment are profoundly impacted by the similarity measure." bioRxiv (2024): 2024-08.
>
> * Williams, Alex H., et al. "Generalized shape metrics on neural representations." Advances in neural information processing systems 34 (2021): 4738-4750.
>
> * Reimers, Nils, and Iryna Gurevych. "Sentence-bert: Sentence embeddings using siamese bert-networks." arXiv preprint arXiv:1908.10084 (2019).
>
> * Dosovitskiy, Alexey. "An image is worth 16x16 words: Transformers for image recognition at scale." arXiv preprint arXiv:2010.11929 (2020).

---

### Meta-Review · Area_Chair_dSXE · 2025-12-29

**Summary:**

The reviewers raised several substantive concerns centering on (i) the empirical justification of the proposed Hierarchical Optimal Transport (HOT) framework beyond existing OT-based and linear baselines, (ii) the necessity and potential bias introduced by hierarchical constraints, (iii) missing or insufficiently justified comparisons to common representational similarity metrics (e.g., RSA, CKA, linear predictivity, global OT), (iv) interpretability claims that were initially qualitative, and (v) computational scalability. Additional concerns included presentation clarity, the scope of experimental validation (e.g., brain areas, architectures), and robustness analyses. One reviewer also expressed limited confidence in assessing originality due to unfamiliarity with the literature, while another review exhibited inconsistencies in scoring and tone that reduced its reliability.

Across the discussion period, the authors provided an extensive and technically detailed rebuttal, including multiple new experiments, clarifications of misconceptions about the method, quantitative additions, and manuscript revisions. Taken together with the strong positive assessments from two reviewers and the partial addressed concerns, the paper was judged to make some contribution to representational alignment, supporting an accept decision. However, authors should explain more about the computational cost, alignment and interpretability, and the downstream application of this paper in the final version, which is much more important.

**Reviewer Concerns:**

The rebuttal addressed the core concerns by clarifying that HOT does not impose hierarchical structure but allows it to emerge from optimization, and by adding or analyzing comparisons with global OT, linear mappings, RSA, and linear predictivity. Interpretability claims were strengthened through clearer definitions and quantitative measures, and robustness was supported with additional analyses (e.g., subsampling and variance reporting). Remaining concerns mainly relate to computational cost and broader experimental scope, which are acknowledged as limitations and do not undermine the main contribution.

**Reviewer Scores:**

Reviewer xsQG was consistently positive and strongly supportive of acceptance. Reviewer 8PNK provided an accept-level assessment with limited confidence, which remains appropriate after the rebuttal. Reviewer 1X33 raised substantive technical concerns that were addressed in the rebuttal and would likely support a weak accept after revisions.

The review from Reviewer K8P6 is difficult to recalibrate due to low reported confidence, lack of updated technical justification after a score downgrade, and issues of tone. Accordingly, limited weight was placed on this score in the final decision.

---

### Decision · Program_Chairs · 2026-01-26

Accept (Poster)